

# Blue Intensity based experiments for reconstructing North Pacific temperatures along the Gulf of Alaska

Rob Wilson[1,2]; Rosanne D'Arrigo[2]; Laia Andreu-Hayles[2]; Rose Oelkers[2]; Greg Wiles[2,3]; Kevin Anchukaitis[4,2] and Nicole Davi[2,5]

[1]School of Earth and Environmental Sciences, University of Saint Andrews, Saint Andrews, UK;

[2]Tree-Ring Laboratory, Lamont-Doherty Earth Observatory, Palisades, NY, USA;

[3]Department of Geology, The College of Wooster, OH, USA;

[4]School of Geography and Development & Laboratory of Tree Ring Research, University of Arizona, Tucson, AZ

USA

[5]William Paterson University, New Jersey, USA.

*Correspondence to*: Rob Wilson (rjsw@st-andrews.ac.uk)

**Abstract:** Climate in the Gulf of Alaska (GOA) reflects large-scale ocean-atmosphere variability of the North Pacific climate system. Ring-width (RW) records from the GOA have yielded a valuable long-term perspective for North Pacific changes on decadal to longer time scales in prior studies, but express a broad winter to late summer seasonal response. Similar to the highly climate-sensitive maximum latewood density (MXD) proxy, the Blue Intensity (BI) parameter has recently been shown to correlate well with year-to-year warm-season temperatures for a number of sites at northern latitudes. Since BI records are much less expensive and labor intensive to generate than MXD, such data hold great potential value for future tree-ring studies in the GOA and other regions at mid-to-high latitudes. Here we highlight the potential for improving tree-ring based reconstructions using combinations of RW and BI-related parameters (latewood BI (LWB) and delta BI (DB)) from an experimental sub-set of samples from eight mountain hemlock (*Tsuga mertensiana*) sites along the GOA. This is the first such study for the hemlock genus using BI data. We find that using either LWB or DB can improve the amount of explained temperature variance by > 10% compared to RW alone although the optimal target season changes to June-September, which may have implications for studying ocean-atmosphere variability in the region. However, one challenge in building these BI records is that resin extraction did not remove colour differences between the heartwood and sapwood, so long term trend biases, expressed as relatively warm temperatures in the 18[th] century, were noted when using the LWB data. Using DB appeared to overcome these trend biases resulting in a reconstruction expressing 18[th]-19[th] century temperatures ca. 0.5°C cooler than the 20[th]/21[st] centuries. This cool period agrees well with previous dendroclimatic studies and the glacial advance record in the region. Continuing BI measurement in the GOA region must focus on sampling more trees per site (> 20) and more sites to overcome site specific factors effecting climate response while sub-fossil material will extend the reflectance records back over 1000 years. DB appears to capture long term secular trends that agree with other proxy archives in the region but great care is needed when implementing different detrending options. Finally, more experimentation is needed to assess the utility of DB for different conifer species around the Northern Hemisphere.

**Keywords:** Blue Intensity, Gulf of Alaska, Tree Rings, Reconstruction, North Pacific**; Short Title**: Gulf of Alaska Blue Intensity Tree-Ring Temperature Reconstruction

## 1. Introduction



The climate of the Gulf of Alaska (GOA) is strongly influenced by the atmosphere-ocean variability of the North
Pacific sector (e.g. the Pacific Decadal Oscillation, Mantua et al. 1997), with profound socioeconomic implications
for the region (Ebbesmeyer et al. 1991). However, the variability of such synoptic climate phenomena is more
strongly expressed in winter. Ring-width (RW) data measured from montane treeline conifer trees in the GOA
region often express a broad seasonal response window (e.g. January-September, Wilson et al. 2007; February-
August, Wiles et al. 2014), which has allowed such data to provide information on cold season synoptic dynamics
for almost two thousand years (Barclay et al. 1999, D'Arrigo et al. 2001, Wiles et al. 2004 and 2014, Wilson et al.
50  2007).


Maximum-latewood density (MXD) measurements have yielded long records of past summer temperatures for many
regions in the northern mid-to-high latitudes (e.g. Schweingruber 1988, Briffa et al. 2002, Anchukaitis et al. 2013,
Schneider et al. 2015), but such records do not yet exist for the GOA. MXD series are particularly desirable as such
records often express stronger coherence with temperatures than RW and result in climate reconstructions with
better skill and spectral fidelity (Anchukaitis et al. 2013, Esper et al. 2015, Wilson et al. 2016). This is partly
because RW chronologies typically exhibit higher autocorrelation and lagged memory effects than MXD (Briffa et
al. 2002; Anchukaitis et al. 2012), but also because RW may potentially integrate other ecological signals (e.g.
disturbance and stand dynamics) which can obscure the climate signal (Rydval et al. 2015). Yet, only two
millennial-length MXD records are currently published for all of northwestern North America (Icefields, British
Columbia (BC), Canada - Luckman and Wilson 2005; Firth River, Alaska - Andreu-Hayles et al. 2011, Anchukaitis
et al. 2013) and no traditionally measured MXD data have been generated to date for the entire GOA. This situation
partly relates to the expensive and labor intensive nature of MXD measurement, but also because the wood of
mountain hemlock (*Tsuga mertensiana*), the dominant conifer species in the GOA, is rather brittle and does not lend
itself well to sample preparation for MXD measurement.

To help meet the need for additional climatically-sensitive density records from northwestern North America, we
present herein an exploration of novel Blue Intensity (BI) parameters measured from scanned images of tree core
samples from the GOA. Minimum latewood blue intensity (LWB) has recently been shown to express strong
similarities to MXD, and is much cheaper and easier to generate (McCarroll et al. 2002; Björklund et al. 2014, 2015;
Rydval et al, 2014; Wilson et al. 2014, 2017). LWB is closely related to MXD as they measure similar wood
properties (combined hemicellulose, cellulose and lignin content related to cell wall thickness), and both are well
correlated with warm-season temperatures (Campbell et al. 2007; Björklund et al. 2014, Rydval et al. 2014, Wilson
et al. 2014). This correspondence between BI and temperature has recently been shown to hold true for several
locations and tree species, including Scots pine (*Pinus sylvestris*) in Scotland, UK (Rydval et al. 2014) and Sweden
(Björklund et al. 2014, 2015), Caucasian fir (*Abies nordmanniana*) in the Northern Caucasus' (Dolgova 2016),
Stone pine (*Pinus cembra*) in Austria (Österreicher et al. 2015; Wilson et al. 2017), Engelmann spruce (*Picea
engelmannii*) from the Canadian Rockies, British Columbia, Canada (Wilson et al. 2014) and our own analyses of
white spruce (*Picea glauca*) in northwestern North America (Andreu-Hayles et al., ms. in prep.). Although BI often

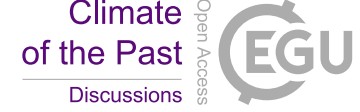

requires larger sample sizes than MXD to improve signal strength (Wilson et al. 2014), this is not a concern due to
the low cost of the method.
The greatest limitation of LWB, however, is that any colour variation that does not represent year-to-year climate-
driven cell wall thickness changes will bias the resultant raw reflectance measurements. For example, some conifer
species (including Scots pine and mountain hemlock) show either a sharp or transitional colour change from the
heartwood to sapwood, which, even after resin extraction using ethanol or acetone, can still impose a systematic
change in reflectance around the heartwood/sapwood transition (Björklund et al. 2014, 2015). Further colour
variations, often seen in dead but preserved snag or sub-fossil wood, can also result in systematic biases when
combined with data measured from living samples (Björklund et al. 2014, 2015; Rydval et al. 2014). Björklund et al.
(2014) proposed a potential solution to the heartwood/sapwood colour bias issue by effectively detrending the LWB
measurements by removing the inherent common colour changes of the earlywood and latewood (i.e. those related
to heartwood/sapwood colour change). This was done by subtracting the LWB value from the maximum blue
reflectance value of the earlywood (EWB) for each year. The resulting new parameter, referred to as delta blue
intensity (hereafter referred to as DB), should theoretically be less biased by such non-climatic related colour
changes. Although Björklund et al. (2014, 2015) presented compelling DB results using Scots pine in Sweden, the
method has not yet been tested elsewhere or on any other species. We hypothesis that DB can only theoretically
work if the inter-annual signal between EWB and LWB is weakly correlated. If the correlation between these two
parameters is high, then the method of deriving DB may remove the specific climate signal of interest.
Finally, although BI based variables hold great promise as an alternative proxy to MXD, another concern is their
potential inability to capture low frequency information related to long term-climate changes. Wilson et al. (2014),
working with Engelmann spruce from British Columbia, which does not express a visual colour difference, urged
caution as both the MXD and LWB parameters were sensitive to different detrending options and there was some
indication that LWB could not capture as much low frequency information as MXD. However, this observation
could not be fully addressed due to the relatively short instrumental record in British Columbia.
In this paper, building upon previous RW based research (Wilson et al. 2007, Wiles et al. 2014), we measure BI
variables (EWB, LWB and DB) from multiple sites in the GOA to evaluate: (a) whether BI can improve on previous
RW-only based reconstruction, and (b) whether meaningful low frequency information can be gleaned from these
data by exploiting the long monthly instrumental temperature records that go back into the mid-19[th] century to
validate secular trends in the TR data.
**2. Methods and Analysis**




BI measurements were made on a sub-set (ca. 15 single tree cores per site) of crossdated core samples collected over
the past few decades from living mountain hemlock (*Tsuga mertensiana* Bong. Carrière) trees located at eight sites
near altitudinal treeline (around ~300-400 meters above sea level) along the GOA (Table 1, Figure 1). Data from
these and additional sites were used previously to create coastal GOA RW based temperature-related reconstructions
(D'Arrigo et al. 2001, Wilson et al. 2007, Wiles et al. 2014).

The tree core samples were immersed in acetone for 72 hours to remove excess resins in the wood (Rydval et al.
2014) and then finely sanded to 1200 grit to remove marks and abrasions prior to scanning. An Epson V850 pro
scanner, using an IT-8 calibration card in conjunction with Silverfast scanning software, was used to scan the
samples at 2400 dpi resolution. EWB and LWB variables were measured using the CooRecorder 8.1 software
(Cybis 2016 - http://www.cybis.se/forfun/dendro/index.htm), which has state-of-the-art capabilities to acquire
accurate reflectance intensity RGB colour measurements from scanned wood samples (see Rydval et al. 2014). DB
values were calculated within CooRecorder by subtracting the LWB values from the EWB values for each year.
Since LWB is negatively correlated to MXD (high density 'dark' latewood = low reflectance), values were inverted
following the method detailed in Rydval et al. (2014) to allow for LWB to be detrended in a similar way to MXD
(see also Wilson et al. 2014). The nature of the DB calculation results in this parameter being positively correlated
with inverted LWB, so these data could also be theoretically detrended in a similar way.

For initial experiments comparing the different tree-ring (TR) variables, the RW, LWB, EWB and DB data were
detrended using fixed 200-year cubic smoothing splines (Cook and Peter 1981) to retain the interannual to decadal
signal and minimize any potential lower frequency biases due to heartwood/sapwood colour changes. These
chronology versions were assessed by (1) signal strength statistics: both common signal (via mean inter-series
correlation – RBAR) and expressed population signal (EPS - Wigley et al. 1984) statistics, (2) between variable
correlation, (3) between site coherence using a rotated principal component analysis (PCA, varimax rotation using
correlation matrices with eigenvectors retained with an eigenvalue > 1.0) and (4) climate response derived by
correlations between regional composite TR variable mean series and the dominant PC scores against monthly and
season variables of temperature (CRU TS 3.24 (Harris et al. 2012): 57-61ºN / 153-134ºW).

The 200-year spline chronology versions were also used to explore calibration (1901-1960) and validation (1961-
1989) based principal component regression reconstruction experiments using the CRU TS data. For the PCA, a
reasonably replicated common period (1792-1989) was used where tree series replication was > 5 trees. All site
chronologies are replicated with > 10 trees from 1792 except for JM and SR (see Table 1) where replication is 6 and
5, respectively. Reconstruction validation was performed using the Pearson's correlation coefficient (r), the
Reduction of Error (RE) and the Coefficient of Efficiency (CE - Cook et al., 1994). Further validation was
performed over the 1850-1900 period using the gridded BEST instrumental data (Rohde et al., 2012), extracted for
the same region as the CRU TS (57-61ºN / 153-134ºW), after these data were scaled to the CRU TS data over the



1901-2015 period. CRU TS and BEST compare well to the original GOA 5-station mean record (Supplementary
Figure 1) used in Wilson et al. (2007) confirming that the gridded products are good representations of the regional
temperature signal. The higher variance of the pre-1950 period in the 5-station mean is related to the fact that
variance stabilization (Frank et al. 2007a) was not performed when this mean series was originally developed
(Wilson et al. 2007) and is therefore likely a less robust measure of GOA temperatures than the gridded products.

Finally, to explore the potential of reconstructing robust low frequency temperature changes in the region, the data
from each of the eight sites were pooled to derive GOA regional composite records for each of the TR variables.
These pooled composite variable datasets, with their greater overall replication, allowed detrending experiments to
be performed to ascertain the sensitivity of the final parameter chronologies to different detrending choices.
Specifically, RW detrending experiments were performed using (1) STD: negative exponential function or negative
or zero slope linear function detrending via division; (2) NEPT: negative exponential function or negative or zero
slope linear function detrending via subtraction after power transformation of the raw RW data (Cook and Peters
1997); (3) RCS: single group regional curve standardization (RCS - Briffa et al., 1996; Esper et al., 2003; Briffa and
Melvin 2008) detrending via division. For each of these three approaches, the 'Signal-Free' (SF - Melvin and Briffa
2008) approach to detrending was also utilized. These different options resulted in 6 different RW composite
chronologies. For LWB and DB, as they theoretically should behave more like MXD, detrending was performed
using only two methods; (1) LINres: negative or zero slope linear function detrending via subtraction; (2) RCSres:
single group RCS detrending via subtraction. As with the RW data, the SF approach was also performed leading to
four chronology variants for both LWB and DB.

**3. Results and Discussion**
**3.1 Common signal within the network**
RW has the strongest common signal with a median overall RBAR of 0.44 (8 site range: 0.33 – 0.49 - Table 1),
whereas LWB and DB both have weaker RBAR values of 0.24. EWB shows the weakest common signal with a
median RBAR from the 8 sites of only 0.12. In order of decreasing between-series common signal, the number of
series needed to attain an EPS of 0.85 are 7 (RW), 18 (LWB and DB), and 41 (EWB) for each TR variable
respectively. Rydval et al. (2014) showed that as the within tree common signal was much weaker for LWB than
RW, the between tree common signal improved more for LWB than RW as multiple radii from the same tree were
measured (i.e. up to 3). For this exploratory analysis, only a single series was measured per tree, and therefore we
hypothesise that the EPS of BI based chronologies would improve markedly, compared to RW, if at least 2 radii
were measured from each tree.

The weak signal strength in EWB compared to RW, LWB and DB is also reflected in the PCA. The leading PC for
RW, LWB and DB explains 59%, 53% and 57% of the overall variance, respectively, while just 39% is explained by
the EWB PC1. In general, the loadings (based on a varimax rotation) of the chronologies on each PC for each



variable are related to the geographical locations across the GOA with PC1 representing the eastern sites and PC2
the western ones (Figures 1 and 2).

**3.2 Seasonal temperature sensitivity**
EWB contains a weak response to summer temperature variability with almost no late summer temperature signal
(Figure 2) although significant correlations (r = ~0.3 - 0.4) are found with May and previous October/November
temperatures (supplementary Figure 2). In agreement with previous work (Wilson et al. 2007; Wiles et al. 2014),
RW correlates well with a broad range of summer seasons, showing positive correlations for nearly all months from
January through to September (Supplementary Figure 2) with June returning the strongest correlation. LWB and
DB, on the other hand, show weaker responses with the late winter/spring months and strongest correlations with
June, July and August (Figure 2). As LWB and DB should express similar growth/climate response properties to
MXD, these observations are not surprising.

There appears to be a geographical difference in response with PC1 (eastern sites) showing stronger seasonal
(Figure 2) and monthly (supplementary Figure 2) correlations with temperature than PC2 (western sites). This
spatial pattern of response is also expressed in the RW data (Figure 2). However, correlations of the individual site
chronologies for each TR variable (Table 2) against June-September temperatures (optimal season for reconstruction
– see later) suggest that there is a degree of variability of the individual sites' response to summer temperatures
across the GOA. As PC2 is weighted more towards the TBB site (see PCA loadings in Figure 2 for RW, LWB and
DB) which correlates weakly with JJAS, it is therefore not surprising that this PC correlates weakly with summer
temperatures. However, the correlation results of the mean composite chronologies (Figure 2) are marginally
stronger than the PC1 results. This suggests that a regional mean composite approach is potentially optimal in the
context of deriving a GOA wide reconstruction which can be extended further back in time using data generated
from sub-fossil samples.

The positive correlation of RW, LWB and DB to summer temperatures (Figure 2 and Table 2) is also reflected in the
inter-correlation between these different variables (Table 3). RW agrees most strongly with DB, followed by LWB.
EWB has the weakest relationship with the other 3 variables. Importantly, the correlation between EWB and LWB is
weak which we hypothesise is the theoretical ideal for the utilization DB to minimize potential heartwood/sapwood
colour change biases (Björklund et al. 2014, 2015). Hereafter, due to the poor signal strength and weak climate
signal, the EWB data alone was not used for further analysis except in the DB calculations.

**3.3 Calibration/validation experiments**
Calibration and validation statistics for various PC regression variable combinations for several summer target
seasons are detailed in Table 4 along with results using the GOA RW composite of Wiles et al. (2014). Firstly,
calibration of Wiles et al. (2014) to the CRU TS 3.24 data (February – August) over the 1901-1989 period ($r^2$ =
0.33) is stronger than the new sample sub-set based RW GOA composite ($r^2$ = 0.27) which also shows a significant





trend in the model residuals. This residual trend possibly reflects the fact that there could be a longer term low
frequency trend missing in the RW data due to the use of 200-year spline detrended chronologies when compared to
the RCS processed version of Wiles et al. (2014). Also, the slightly weaker results of the new RW data likely reflect
generally lower replication in the current study compared to Wiles et al. (2014).
The strongest calibration $_ar^2$ values for each BI parameter over the 1901-1960 period are 0.49 and 0.47 for LWB and
DB respectively for the JJA season although DB fails validation with negative RE and CE values over the 1961-
1989 period. Minimal model improvement is gained by including RW data. RW+LWB calibrates best ($_ar^2$ = 0.49)
with JJA while RW+DB explains more temperature variance for MJJAS ($_ar^2$ = 0.51). However, in both cases,
validation RE and CE are negative. Focussing on the full period (1901-1989) calibration, strongest results are found
for the JJAS season for all parameters options (except Feb-Aug for RW) with $_ar^2$ values of 0.27 (RW), 0.43 (LWB),
0.38 (DB), 0.38 (RW+LWB) and 0.39 (RW+DB) with no 1st order autocorrelation noted for any version. Only the
RW+DB version, however, shows no significant linear trend in the model residuals. The full period (1901-1989)
calibrated reconstructions (Table 4) for each of the variable options are presented in Figure 3 along with independent
validation (1850-1900) with the BEST gridded data. All parameter iterations fail validation (negative CE values)
except for RW+DB which returns positive RE (0.57) and CE (0.19) values. Overall, using this subset of samples
from these 8 sites, the calibration results (Table 4 and Figure 3) indicate that BI based parameters explain more
temperature variance than using RW alone. However, assessing the fidelity of the resultant reconstructions appears
sensitive to the periods of calibration and validation used and as the chronologies were limited in the frequency
domain by using a fixed 200-year spline detrending option, it is not clear which of these parameters best represent
longer term secular change.
The large-scale climate signal expressed by these data is illustrated by comparing the RW+DB JJAS reconstruction
with gridded land/sea HadCRUT4 (Morice et al. 2012; Cowtan and Way 2014 – Figure 4a) and land only CRU TS
3.24 (Harris et al. 2012 – Figure 4b) temperatures for the GOA and North Pacific sector. Although the spatial
correlations are stronger towards Juneau and Sitka (see Figure 1 for locations) in the east of the region it is clear that
these new data represent very well the temperature variability of the wider GOA region and North Pacific.
Continued measurement of BI based parameters from sub-fossil samples taken from each end of the GOA will allow
long term summer temperature variability to be derived for at least the last millennium which will complement the
long RW based temperature reconstructions expressing a broader seasonal window (Wilson et al. 2007; Wiles et al.
257  2014).

**3.4 Potential low frequency bias**
The main potential limitation to the use of BI based TR variables such as LWB is concerned with low frequency
trend biases related to wood colour change. Mountain hemlock, in general, shows darker heartwood and lighter
sapwood, a colour change which resin extraction appears to only minimise but not entirely remove. Also, this colour
change is not a sharp transition and is expressed in raw EWB and LWB measurements as a steady increase in

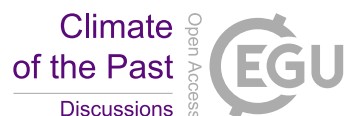

reflectance intensity. Non-detrended mean composite chronologies of EWB and LWB for the whole GOA region
(Figure 5) clearly show the impact of the heartwood/sapwood colour change with increasing intensity values through
time (see also Supplementary Figure S3 for a single tree example), especially since the late 18[th] century. However,
MXD generally shows a linear decreasing trend with increasing cambial age (Esper et al. 2012). If LWB is indeed a
comparable (but inverted) TR variable to MXD as a measure of latewood anatomical density properties, then we
would expect, therefore, an increasing trend in raw LWB values. Figure 5 therefore poses a potential "mixed-signal"
conundrum as the observed trend in the GOA raw mean LWB composite will incorporate both the true age-related
trend of changing latewood density and the heartwood/sapwood colour change bias. Although using DB can
theoretically overcome these colour bias issues, it has not been explored in any detail beyond the original concept
papers (Björklund et al. 2014, 2015). The mean DB non-detrended GOA chronology (Figure 5) expresses minimal
long term trends which could suggest that the colour change bias has been removed or at least minimised.
Mean cambial age-aligned curves of the LWB and DB data show very distinct trends (Supplementary Figure 4).
LWB appears to show a general linear increase in values – a trend that would be expected if LWB indeed does
reflect similar wood properties (inversely) to MXD. DB, however, has a more complex mean growth curve and
shows an initial increasing juvenile trend for ~50 years, a period of stabilisation and then a decreasing trend from
about ~200 to 300 years. These different age-aligned curves highlight that different detrending options may well be
needed for these different TR variables.
A range of credible methodological choice options for detrending the RW, LWB and DB GOA regional composite
data are presented in Figure 6. The outcome for the RW data appears extremely consistent even when using STD vs
RCS based methods. However, the LWB and DB chronologies are extremely sensitive to the detrending method
used. Compared to RW and DB, all LWB chronology variants show above zero z-score values in the 17[th] century,
which likely reflects the low reflectance bias of the darker heartwood compared to the sapwood because the LWB
data have been inverted. The RCS versions appear particular inflated and as LWB is positively correlated with
summer temperatures (Figure 2), this would result in markedly warm temperature estimates during the LIA
compared to the 20[th] century which is at odds with previous GOA dendroclimatic analyses (Wiles et al. 2014) and
the geomorphological record, which indicates substantial glacial advance from the 17[th] to 19[th] centuries (Wiles et al.,
2004; Solomina et al. 2016). RCS can impart significant low frequency bias when the assumptions and requirements
of the method are not met (Melvin and Briffa 2014; Anchukaitis et al. 2013) and as the GOA composite utilises only
living trees this is a far from optimal sample design for this detrending method. For DB, the LINsf version deviates
markedly from LINres, RCSres and RCSsf variants with very low values (< -6 standard deviation from 1901-1989
mean) before 1700 followed by a strong linear increase until present. A similar observation was noted in Wilson et
al. (2014) where signal free detrending of LWB and especially MXD resulted in much cooler LIA conditions than
other detrending approaches.



**3.5 JJAS GOA summer temperatures back to 1600**

The long GOA instrumental record allows for additional assessment of how different reflectance based chronology variants track temperatures back through time. Using the extended BI based regional composite records, further reconstruction experiments against the JJAS season were performed using LWB and DB separately by calibrating against JJAS CRU TS3.24 (1901-2010) and separately validating using the BEST data (1850-1900). For the LWB data, RCSres and RCSsf calibrated poorly (Table 5: $r^2 = 0.07$ and 0.05 respectively) with negative CE values over the 1850-1900 period. The LINres and LINsf version, however, explained 41% of the temperature variance and validated reasonably well with positive RE and CE values. Significant 1$^{st}$-order autocorrelation (DW range 1.28 to 1.37) and linear trends (LINr range 0.36 to 0.48) was however noted for all model residuals. The DB chronology variants on the whole performed better than their LWB counterparts, with RCSsf (RCSres) calibrating most strongly ($r^2 = 0.43$ (0.40)) and validating well (RE = 0.48 (0.50), CE = 0.15 (0.18)). The residuals for both versions show no 1$^{st}$-order autocorrelation although a significant linear trend is still however observed.

The best reconstructions using LWB (LINres) and DB (RCSsf), identified using the calibration and verification results (Table 5), represent quite different histories of past GOA temperatures (Figure 7). Specifically, the LWB (LINres) reconstruction expresses temperature estimates from the late 17$^{th}$ to mid-19$^{th}$ century warmer than the 1961-1990 mean, while the DB (RCSsf) reconstruction exhibits generally cooler conditions. Both reconstructions explain a similar amount of summer temperature variance and validate well (Table 5) and from comparison to the instrumental data alone, one cannot objectively choose which of the two is most robust although there are arguably less problems with the model residuals using the DB data. Wilson et al. (2014) highlighted the difficulties of relying solely on the instrumental data to validate the long-term trend in any reconstruction. Moreover, there could be unknown homogeneity issues in early instrumental data series which are difficult to identify which would impact calibration and validation (see Frank et al. 2007b). Therefore, alternative sources of relevant information are needed for further validation. As the geomorphological record in the region suggests a prolonged period of glacial advance occurred in the GOA up to the early 20$^{th}$ century (Wiles et al., 2004; Solomina et al. 2016) when a substantial retreat started, we hypothesize that the pre-1900 period must therefore have been cooler. This would suggest that the DB based reconstruction is likely more representative of past GOA temperatures than the LWB driven one.

Figure 8 presents the RW + DB principal component reconstruction (Figure 3), the DB (RCSsf) extended reconstruction (Figure 6) and the Wiles et al. (2014) RW based reconstruction and compares them to the GOA regional glacial advance record (Wiles et al., 2004; Solomina et al. 2016). The TR reconstructions demonstrate centennial and multi-decadal agreement, although the extended DB reconstruction has a smaller amplitude of temperature change between the LIA period and the 20$^{th}$ century. Overall, temperatures in the GOA region were below the 1961-90 norm throughout most of the LIA with temperatures only rising to substantially higher values in the early 20$^{th}$ century. The coldest decadal periods are centred around the 1700s, 1750s, and 1810s. The glacial advance record clearly shows periods of advance through the LIA, peaking at the end of the 19$^{th}$ century. Despite the use of 200-year spline detrended chronologies, the RW+DB reconstruction has a similar amplitude change to the





Wiles et al. (2014) record, which was derived from RCS processed RW data. It should however be noted also that
this RW based reconstruction was calibrated against Feb-August temperatures which has a greater increasing
temperature trend (0.81°C/century vs 0.62°C/century) and higher variance (0.79 vs 0.41) than JJAS (calculated using
BEST data from 1850-2015), which will influence the amplitude of the reconstructions (Esper et al. 2005).

**4. Conclusions**
We have described a set of experimental temperature reconstructions based on RW, LWB and DB data measured
from eight tree-ring sites along the Gulf of Alaska. Focusing on these data sets, the results demonstrate that
inclusion of BI based variables can significantly improve the calibrated variance explained using RW alone by more
than 10%.

RW, LWB and DB are strongly correlated with each other (Table 3) but the inclusion of LWB or DB shifts the
calibrated signal from a broad (February-August, Wiles et al. 2014) season using RW alone to a late summer (JJAS)
season. The influence of late winter and early spring temperatures on RW suggest that this variable may, in fact, still
be the more optimal variable for studying important synoptic phenomena such as north Pacific variability, which
dominates in the winter/spring months (Wilson et al. 2007).

The LWB data, for mountain hemlock, despite calibrating and validating in a similar way to DB, are clearly affected
by heartwood/sapwood colour differences which impart a trend bias in the resultant chronologies and
reconstructions (Figure 6 and 7). However, this bias may not necessarily always occur for other species showing a
heartwood/sapwood colour change which could be removed through traditional resin extraction methods. For the
first time since the original concept papers by Björklund et al. (2014, 2015), we have experimented with the DB
variable and the resulting reconstruction agrees well with a previous RW based reconstruction (Wiles et al. 2014)
and the glacial advance record (Wiles et al., 2004; Solomina et al. 2016) for the region.

The analyses presented herein must be viewed as a series of experiments to inform future dendroclimatologists of
possible methodological strategies that need to be considered for improving TR based reconstructions using blue
reflectance based variables. Specific to the GOA region, but likely relevant to other regions and species, we
therefore detail the following recommendations:
• Although MXD typically has a higher expressed signal strength and climate responses than RW (Wilson
and Luckman 2003), signal strength in LWB and DB in GOA hemlock is weaker than RW, so replication
needs to be substantially increased (ideally > 20 trees – Table 1) to allow the development of robust
chronologies. Rydval et al. (2014) also showed that substantial improvement in LWB signal strength could
be gained by measuring 2 or even 3 radii per tree. Additional assessments of signal strength should be
conducted as new species and sites are analysed using BI methods.
• For conifer species with a clear colour difference between the heartwood and sapwood, LWB may likely
always express biased long-term trends. The DB variable could potentially minimize this effect as shown



here, but more experimentation with this parameter is needed before it can be commonly used as a solution
to the LWB colour bias problem. Rydval et al. (2017) overcame the heartwood/sapwood colour bias by
utilising a band-pass approach to calibration, where LWB drove the decadal and high frequency fraction of
the Scottish temperature reconstruction, while RW drove the low frequency variability. This approach
however assumes that (1) RW is predominantly controlled by summer temperatures (not necessarily the
case in the GOA) and (2) meaningful longer-term secular information can be gleaned from RW data, which
may not always be the case (Esper et al. 2012).
• The results presented herein highlighted substantial sensitivity of the final chronologies to varying
methodological detrending approaches. Much more exploration of the impact of different detrending
choices is needed for dendroclimatology as a whole. Locations with long instrumental records may help
identify more optimal detrending options but care is needed, as it cannot be assumed that the quality of 19th
century data is comparable to late 20th/early 21st century data. Utilizing other proxy observations of past
climate (e.g. in this case the glacial record) may help further constrain TR estimates of past climate
especially when different chronology variants (that validate well) portray quite different past temperature
histories.
**Acknowledgments.** This is a contribution to the PAGES 2k Network [through the Arctic/North America-2k
working groups]. Past Global Changes (PAGES) is supported by the US and Swiss National Science Foundations.
We also gratefully acknowledge the National Science Foundation's Paleoclimatic Perspectives on Climatic Change
(P2C2) Grant Nos.  AGS 1159430, AGS 1502186 and AGS1502150.    Lamont-Doherty Earth Observatory
Contribution No. 0000.

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



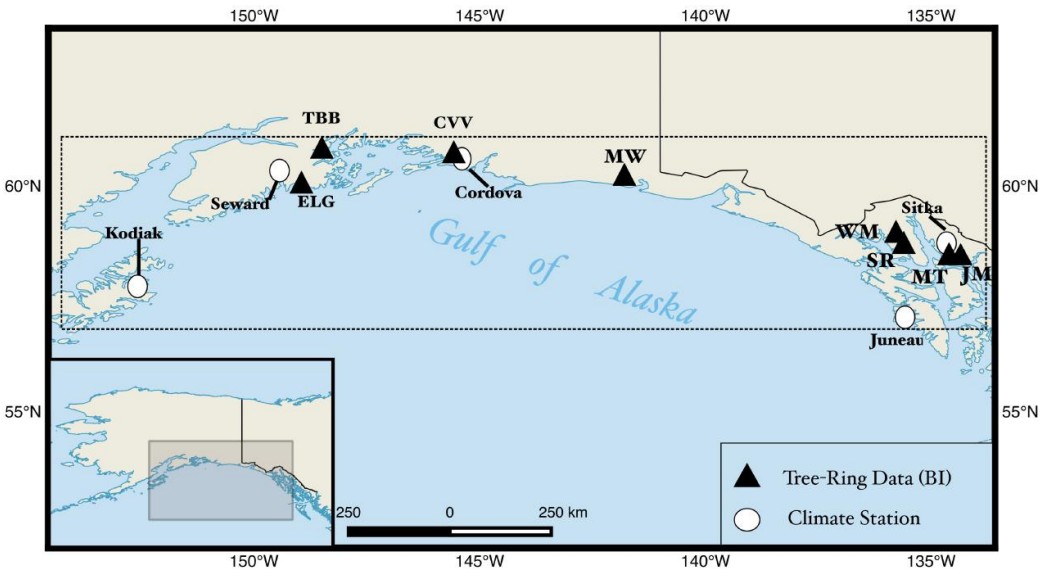

**Figure 1:** Location map of the eight GOA tree-ring sites used in this study (Table 1). Also indicated (dashed
line box) is the domain (57-61°N / 153-134°W) of the gridded data (CRU TS 3.24, Harris et al. 2012; BEST,
Rohde et al., 2012) used for calibration and the five coastal GOA temperature stations used in the original 5-
station mean series (Wilson et al. 2007 – see supplementary Figure 1).



**Figure 2: Left:** Correlation response function analysis (1901-1989) using CRU TS3.24 mean temperatures
with each tree-ring variable (RW = ring-width; EWB = early wood maximum blue intensity; LWB = inverted
latewood minimum blue intensity; DB – Delta Blue). The bars represent correlations with seasonal
temperature for each principal component (PC) score and the simple GOA mean composite. Also for RW,
correlations are shown for the Wiles et al. (2014) RW based RCS reconstruction. Horizontal line denotes the
95% confidence limit. Correlations against individual months are presented in Supplementary Figure 2.
**Right:** Varimax rotation principal component analysis results showing loadings of each chronology on each
PC with an eigenvalue > 1.0. % values denote the explained amount of variance each PC explains of the
original data input matrix.




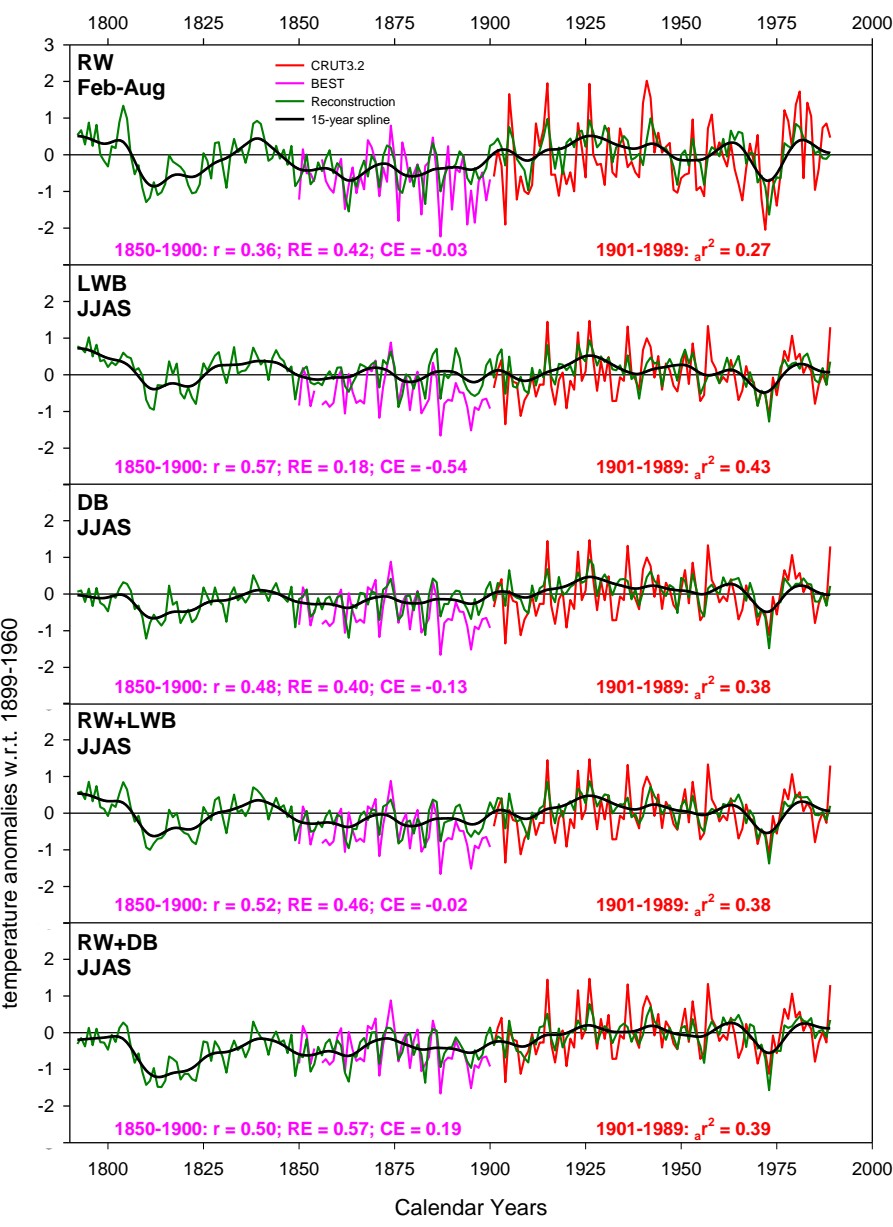

**Figure 3:** Illustration of the various PC regression experiments performed herein, with each reconstruction
model compared against the June-September (Table 3). Feb-August is shown for RW as that was the
reconstructed season in Wiles et al. (2014). Full period calibration is performed on the 1901-1989 period
(Table 3 – CRU TS 3.24) while validation (Pearson's correlation coefficient (r), Reduction of Error (RE) and



Coefficient of Efficiency (CE)) is undertaken over 1850-1900 using the BEST gridded data after those data
were scaled to the CRU TS 3.24 data over the 1901-1989 period.

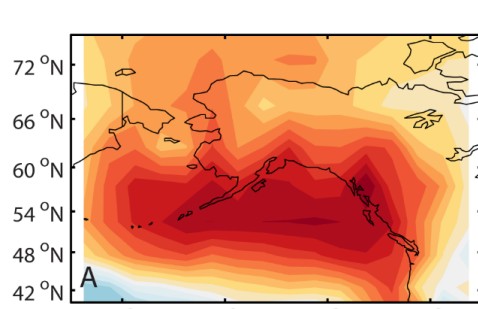
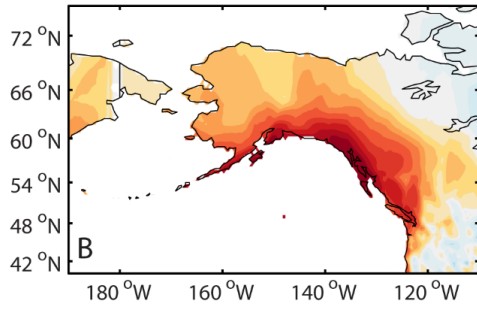
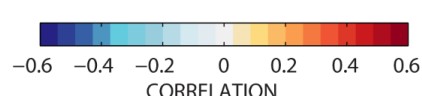


**Figure 4**: Spatial correlation (1901-1989) fields comparing the RW+DB GOA JJAS temperature reconstruction
with larger-scale temperatures. **A:** for HADCRUT4 land/SST (Morice et al. 2012; Cowtan and Way 2014); **B:**
for CRU TS3.24 land temperatures (Harris et al. 2012).



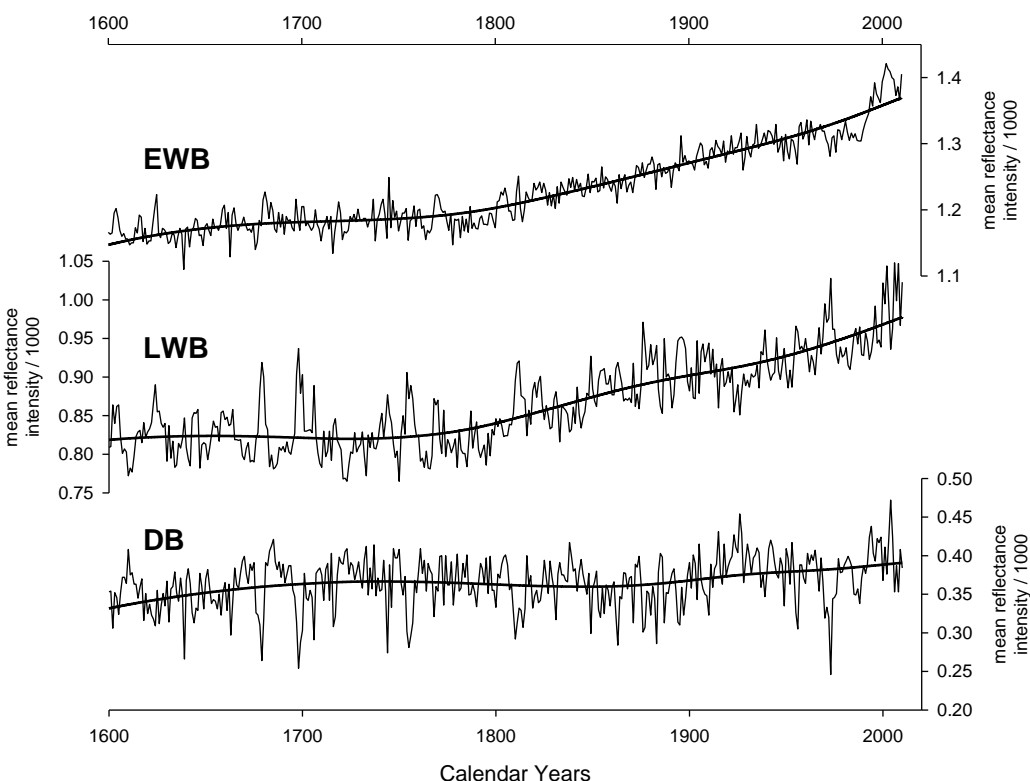



**Figure 5:** Mean non-detrended GOA wide composite chronologies since 1600 for EWB, LWB and DB. The LWB data have
not been inverted for this figure.



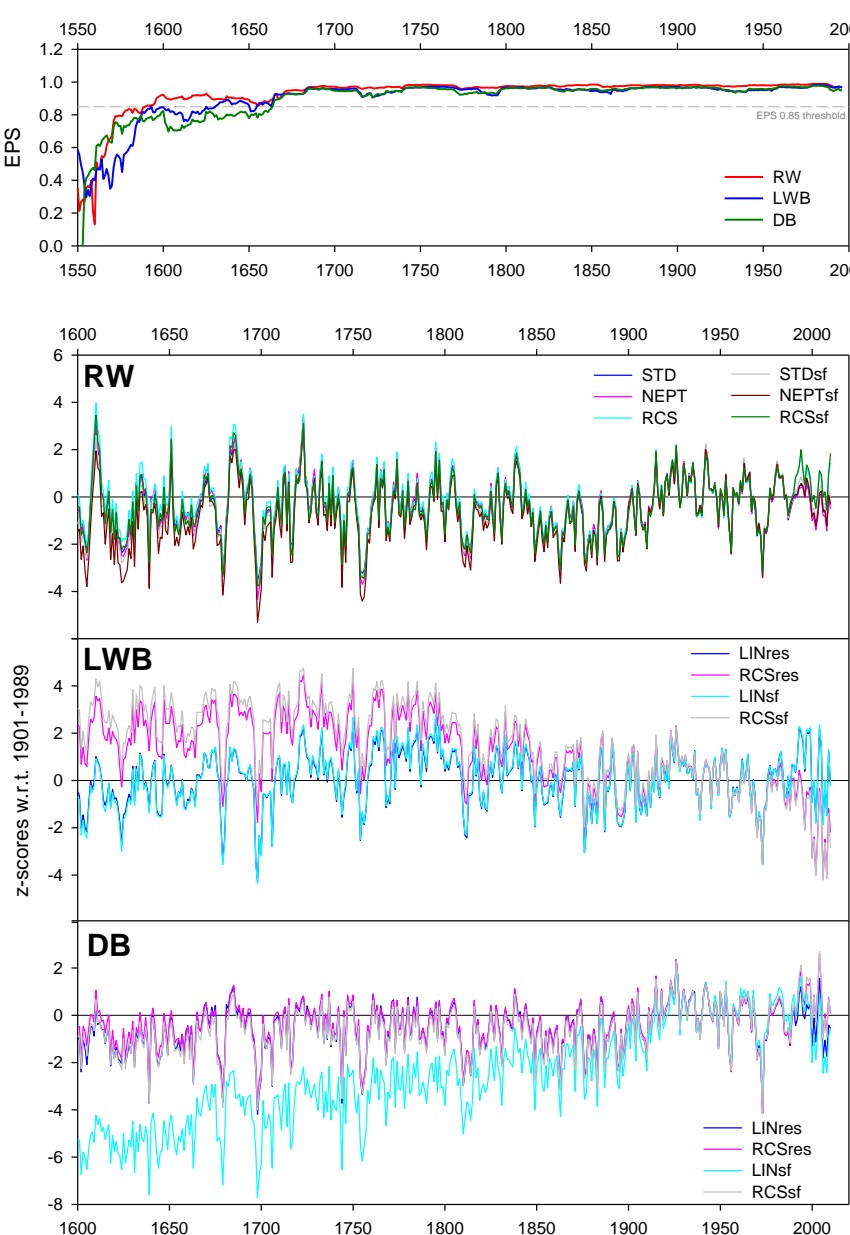

**Figure 6:** Detrending experiments for each TR variable using the full GOA regional composite (data from all 8 sites). Upper panel is 31-year moving EPS plots for RW, LWB and DB using 200-yr spline detrending. Low plots present chronology variants from 1600-2010. For RW - STD = negative exponential detrending (ratio) or regression function of zero or negative slope; NEPT = as STD but raw data have been power transformed and detrended via subtraction; RCS = single group RCS detrending (ratio); STDsf, NEPTsf, RCSsf = as previous three options but using signal free detrending. For LWB and DB – LINres – detrending via subtraction using



linear functions (negative or zero slope); RCSres = as RCS above but detrending via subtraction; LINsf and
RCSsf = as with LINres and RCSres but with signal free detrending.

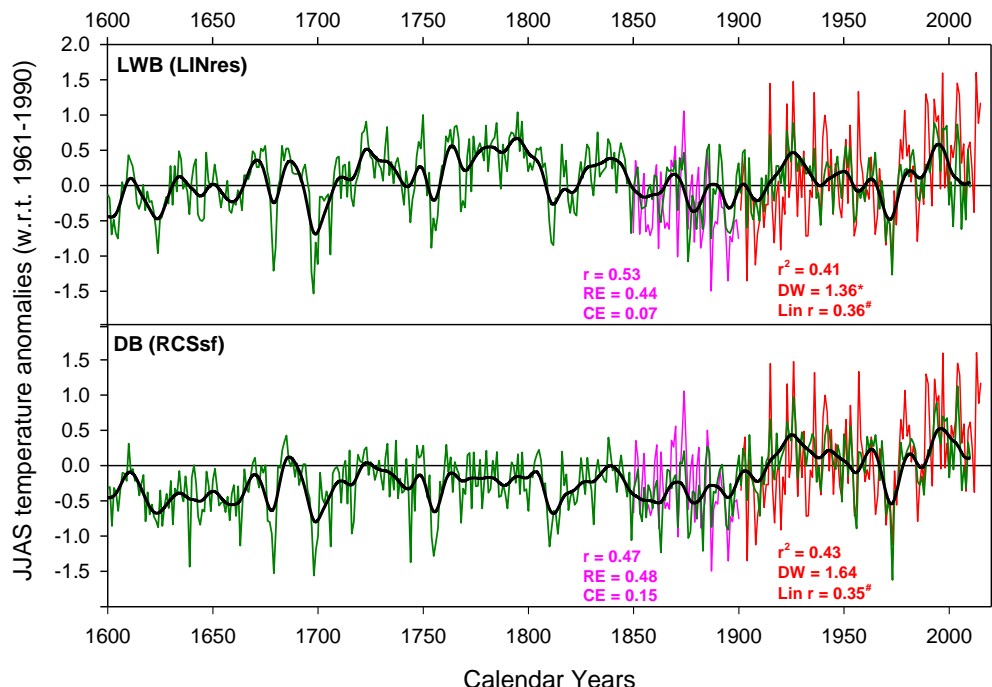



**Figure 7:** Extended reconstruction tests using LWB and DB. 1901-2010 period calibration uses CRU TS3.24
data while validation is performed using BEST data over the 1850-1900 period. * denotes significant 1st order
autocorrelation in model residuals; # denotes a significant linear trend in the model residuals.





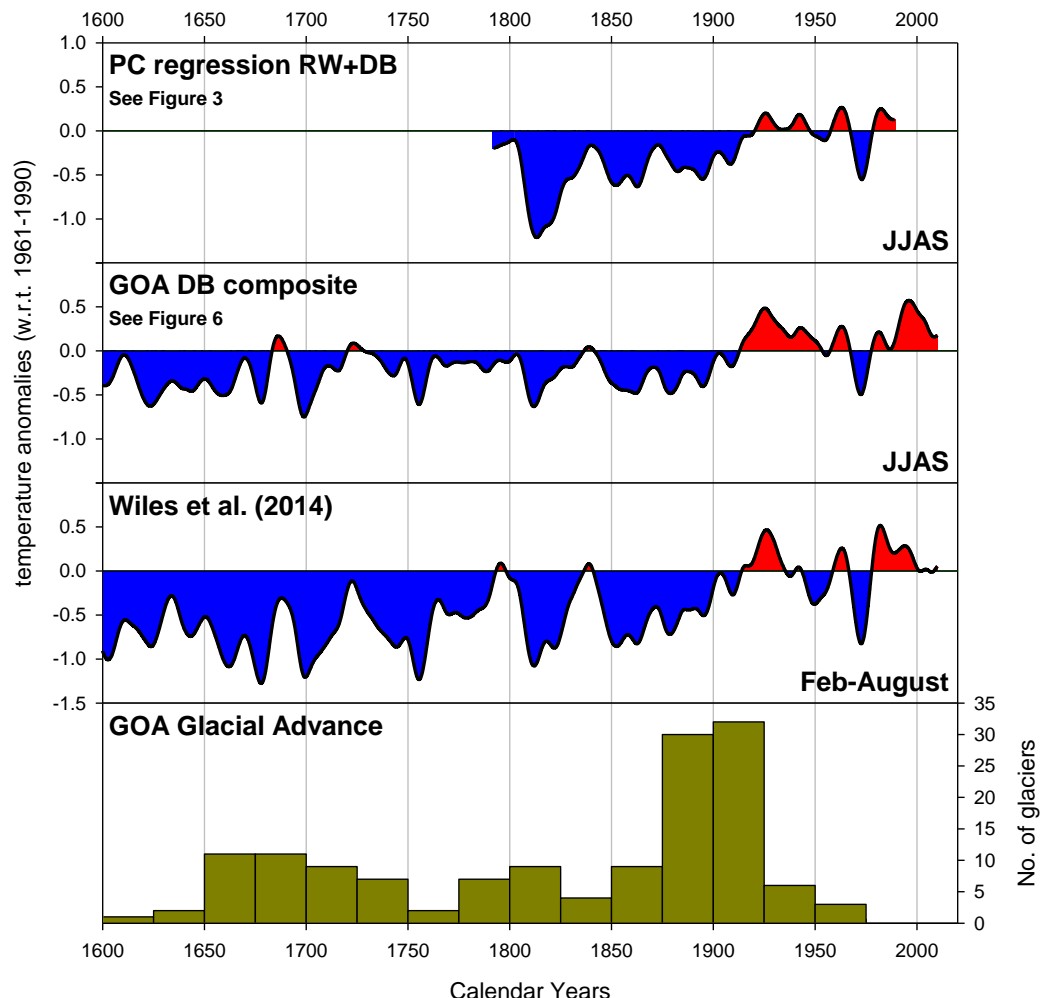

**Figure 8:** Comparison of GOA reconstruction variants using DB and RW with Wiles et al. (2014). The lower panel presents a histogram of glacial advance in the GOA region (Wiles et al., 2004; Solomina et al. 2016).





| Site Name | Timespan | No. of series | Period n > 5 | MSL | RW RBAR | EWB RBAR | LWB RBAR | DB RBAR | N-RW EPS | N-EWB EPS | N-LWB EPS | N-DB EPS |
|---|---|---|---|---|---|---|---|---|---|---|---|---|
| Cordova Eyak Mtn (CVV) | 1573-1992 | 17 | 1672-1992 | 280.6 | 0.46 | 0.15 | 0.32 | 0.29 | 6.5 | 32.1 | 12.0 | 14.2 |
| Juneau Mtn (JM) | 1558-1998 | 17 | 1604-1998 | 238.5 | 0.35 | 0.15 | 0.24 | 0.25 | 10.7 | 32.9 | 17.7 | 17.3 |
| McGinnis (MT) | 1485-1999 | 15 | 1584-1999 | 363.5 | 0.47 | 0.11 | 0.24 | 0.24 | 6.4 | 46.3 | 17.6 | 18.0 |
| Miners Well (MW) | 1479-1994 | 13 | 1640-1995 | 324.0 | 0.49 | 0.05 | 0.33 | 0.14 | 5.8 | 120.3 | 11.8 | 36.2 |
| Son of Repeater (SR) | 1713-2009 | 10 | 1792-2007 | 216.7 | 0.33 | 0.12 | 0.17 | 0.25 | 11.3 | 43.5 | 27.5 | 17.4 |
| Wright Mtn (WM) | 1610-2010 | 17 | 1738-2010 | 234.2 | 0.45 | 0.06 | 0.25 | 0.17 | 6.9 | 84.0 | 17.1 | 27.9 |
| Ellsworth (ELG) | 1636-1991 | 18 | 1750-1990 | 218.5 | 0.40 | 0.14 | 0.24 | 0.29 | 8.6 | 35.7 | 17.7 | 14.2 |
| Tebenkof (TBB) | 1357-1990 | 15 | 1605-1990 | 339.2 | 0.43 | 0.13 | 0.20 | 0.22 | 7.6 | 37.9 | 22.4 | 19.6 |
| | | | median | 259.6 | 0.44 | 0.12 | 0.24 | 0.24 | 7.2 | 40.7 | 17.6 | 17.7 |

**Table 1:** Metadata information for the eight GOA sites used in the study. All PCA and related analyses were performed on the 1792-1990 period for which there is replication for all eight sites of at least five series. Tree-ring data were detrended using a 200-year spline for these signal strength analyses. The final 4 columns denote the number of series needed to attain an EPS of 0.85 (Wigley et al. 1984).

| RW | | | | | | | | |
|---|---|---|---|---|---|---|---|---|
| | JM | MT | SR | WM | MW | CVV | TBB | ELG |
| **1850-1900** | 0.45 | 0.22 | 0.49 | 0.22 | 0.27 | 0.30 | 0.30 | 0.38 |
| **1901-1990** | 0.49 | 0.26 | 0.42 | 0.41 | 0.36 | 0.41 | 0.20 | 0.34 |
| **1850-1990** | 0.50 | 0.37 | 0.47 | 0.43 | 0.39 | 0.47 | 0.33 | 0.39 |
| | | | | | | | | |
| LWB | | | | | | | | |
| | JM | MT | SR | WM | MW | CVV | TBB | ELG |
| **1850-1900** | 0.45 | 0.29 | 0.48 | 0.38 | 0.49 | 0.40 | 0.25 | 0.46 |
| **1901-1990** | 0.52 | 0.39 | 0.64 | 0.58 | 0.46 | 0.45 | 0.28 | 0.41 |
| **1850-1990** | 0.37 | 0.32 | 0.55 | 0.46 | 0.53 | 0.51 | 0.33 | 0.44 |
| | | | | | | | | |
| DB | | | | | | | | |
| | JM | MT | SR | WM | MW | CVV | TBB | ELG |
| **1850-1900** | 0.45 | 0.29 | 0.49 | 0.23 | 0.35 | 0.40 | 0.37 | 0.48 |
| **1901-1990** | 0.57 | 0.45 | 0.58 | 0.47 | 0.48 | 0.50 | 0.23 | 0.39 |
| **1850-1990** | 0.53 | 0.44 | 0.48 | 0.38 | 0.44 | 0.52 | 0.34 | 0.44 |

**Table 2:** Correlations (1850-1900, 1901-1990 and 1850-1990) for each site RW, LWB and DB chronology against JJAS temperatures. EWB correlations are not shown. The sites are ordered from east to west (see Figure 1).

| mean r | EWB | LWB | DB |
|---|---|---|---|
| **RW** | 0.27 | 0.68 | 0.81 |
| **EWB** | | -0.23 | 0.36 |
| **LWB** | | | 0.80 |

**Table 3:** Correlation matrix between the different tree-ring variable chronologies (200-year spline detrended. These values represent the averages for between TR variable correlations performed separately for each site.




| Wiles14 | season | series entered | 1901-1960 Calibration | | 1961-1989 Validation | | | 1901-1989 Full Calibration + Residuals | | | |
|---|---|---|---|---|---|---|---|---|---|---|---|
| | | | r | r2 | r | RE | CE | r | r2 | DW | Linr |
| | MJJA | Wiles2014 | 0.60 | 0.36 | 0.48 | 0.11 | 0.10 | 0.55 | 0.30 | 1.62 | 0.15 |
| | MJJAS | Wiles2014 | 0.55 | 0.30 | 0.53 | 0.21 | 0.20 | 0.53 | 0.28 | 1.72 | 0.17 |
| | JJA | Wiles2014 | 0.58 | 0.34 | 0.47 | 0.06 | 0.05 | 0.53 | 0.28 | 1.75 | 0.17 |
| | JJAS | Wiles2014 | 0.52 | 0.27 | 0.53 | 0.20 | 0.20 | 0.51 | 0.26 | 1.77 | 0.18 |
| | Feb-Aug | Wiles2014 | 0.60 | 0.36 | 0.54 | 0.23 | 0.23 | 0.57 | 0.33 | 1.78 | 0.17 |
| | | | | | | | | | | | |
| RW | season | PCs entered | r | ar2 | r | RE | CE | r | ar2 | DW | Linr |
| | MJJA | 1, 2 | 0.60 | 0.34 | 0.40 | 0.03 | 0.03 | 0.52 | 0.26 | 1.60 | 0.21 |
| | MJJAS | 1, 2 | 0.56 | 0.29 | 0.46 | 0.15 | 0.14 | 0.52 | 0.25 | 1.70 | 0.23 |
| | JJA | 1, 2 | 0.58 | 0.31 | 0.42 | -0.01 | -0.02 | 0.51 | 0.24 | 1.74 | 0.23 |
| | JJAS | 2, 1 | 0.53 | 0.26 | 0.49 | 0.16 | 0.15 | 0.51 | 0.24 | 1.76 | 0.24 |
| | Feb-Aug | 2, 1 | 0.60 | 0.36 | 0.46 | 0.08 | 0.08 | 0.54 | 0.27 | 1.75 | 0.23 |
| | | | | | | | | | | | |
| LWB | season | PCs entered | r | ar2 | r | RE | CE | r | ar2 | DW | Linr |
| | MJJA | 1, 2 | 0.63 | 0.38 | 0.49 | 0.15 | 0.14 | 0.57 | 0.31 | 1.39 | 0.20 |
| | MJJAS | 1, 2 | 0.63 | 0.37 | 0.55 | 0.23 | 0.22 | 0.59 | 0.34 | 1.50 | 0.23 |
| | JJA | 1, 2 | 0.71 | 0.49 | 0.58 | 0.16 | 0.15 | 0.66 | 0.42 | 1.45 | 0.25 |
| | JJAS | 1, 2 | 0.69 | 0.46 | 0.64 | 0.27 | 0.27 | 0.66 | 0.43 | 1.51 | 0.27 |
| | | | | | | | | | | | |
| DB | season | PCs entered | r | ar2 | r | RE | CE | r | ar2 | DW | Linr |
| | MJJA | 1, 2 | 0.69 | 0.45 | 0.43 | -0.01 | -0.02 | 0.59 | 0.33 | 1.55 | 0.24 |
| | MJJAS | 1, 2 | 0.67 | 0.43 | 0.50 | 0.09 | 0.08 | 0.61 | 0.37 | 1.68 | 0.26 |
| | JJA | 1, 2 | 0.70 | 0.47 | 0.50 | -0.05 | -0.05 | 0.62 | 0.36 | 1.65 | 0.26 |
| | JJAS | 1, 2 | 0.68 | 0.44 | 0.58 | 0.11 | 0.11 | 0.63 | 0.38 | 1.72 | 0.29 |
| | | | | | | | | | | | |
| RW + LWB | season | PCs entered | r | ar2 | r | RE | CE | r | ar2 | DW | Linr |
| | MJJA | 1, 2 | 0.69 | 0.45 | 0.46 | 0.03 | 0.03 | 0.60 | 0.34 | 1.67 | 0.18 |
| | MJJAS | 1, 2 | 0.66 | 0.43 | 0.51 | 0.13 | 0.12 | 0.62 | 0.37 | 1.87 | 0.20 |
| | JJA | 1, 2, 3 | 0.72 | 0.49 | 0.52 | -0.07 | -0.08 | 0.63 | 0.37 | 1.56 | 0.26 |
| | JJAS | 1, 2, 3 | 0.68 | 0.43 | 0.59 | 0.12 | 0.12 | 0.63 | 0.38 | 1.63 | 0.28 |
| | | | | | | | | | | | |
| RW + DB | season | PCs entered | r | ar2 | r | RE | CE | r | ar2 | DW | Linr |
| | MJJA | 1, 2 | 0.71 | 0.49 | 0.44 | -0.16 | -0.16 | 0.62 | 0.36 | 1.69 | 0.04 |
| | MJJAS | 1, 2 | 0.72 | 0.51 | 0.49 | -0.14 | -0.15 | 0.61 | 0.36 | 1.78 | -0.11 |
| | JJA | 1, 2, 3 | 0.72 | 0.50 | 0.49 | -0.15 | -0.15 | 0.56 | 0.32 | 1.89 | -0.12 |
| | JJAS | 1, 2 | 0.71 | 0.49 | 0.52 | -0.18 | -0.18 | 0.64 | 0.39 | 1.84 | 0.05 |

**Table 4:** Calibration experiments for four dominant seasons (see Figure 2). Initial calibration (using CRU TS 3.24) was
made over 1901-1960 and validation over 1961-1989. Full calibration (1901-1989) was also performed to allow for
residual tests and extra validation using BEST (1850-1990 – see Figure 2). Shaded results do not pass significance. r =
Pearson's correlation coefficient; r2 = coefficient of determination; ar2 = r2 adjusted for the number of predictors in the
model; RE = Reduction of Error; CE = Coefficient of Efficiency; DW = Durbin-Watson test for residual autocorrelation; LINr
= linear trend of the residuals.



| LWB | 1901-2010 Calibration | | | | | 1850-1900 Validation | | |
|---|---|---|---|---|---|---|---|---|
| | series entered | r | r2 | DW | LINr | r | RE | CE |
| | LINres | 0.64 | 0.41 | 1.36 | 0.36 | 0.53 | 0.44 | 0.07 |
| | RCSres | 0.26 | 0.07 | 1.28 | 0.48 | 0.56 | 0.01 | -0.64 |
| | LINsf | 0.64 | 0.41 | 1.37 | 0.36 | 0.53 | 0.43 | 0.06 |
| | RCSsf | 0.21 | 0.05 | 1.32 | 0.46 | 0.56 | -0.05 | -0.73 |

| DB | 1901-2010 Calibration | | | | | 1850-1900 Validation | | |
|---|---|---|---|---|---|---|---|---|
| | series entered | r | r2 | DW | LINr | r | RE | CE |
| | LINres | 0.55 | 0.31 | 1.37 | 0.50 | 0.50 | 0.52 | 0.21 |
| | RCSres | 0.64 | 0.40 | 1.59 | 0.40 | 0.48 | 0.50 | 0.18 |
| | LINSF | 0.54 | 0.29 | 1.35 | 0.38 | 0.43 | 0.40 | 0.00 |
| | RCSsf | 0.65 | 0.43 | 1.64 | 0.35 | 0.47 | 0.48 | 0.15 |


**Table 5:** Extended reconstruction Calibration experiments using different chronology versions (Figure 6) of LWB and DB.
Shaded results do not pass significance.