# Peer review of "Blue Intensity based experiments for reconstructing North Pacific temperatures along the Gulf of Alaska"

_Climate of the Past, 2017_

## Referee Comment (RC1) · J. Björklund (Referee) · 24 Apr 2017

General comments:

The manuscript experiments with the less expensive surrogate to Xray-density MXD, namely the flatbed-scanner BI method, latewood BI parameter, for a previously unexplored species and part of the world. The authors are particularly interested in the lower frequency fidelity of LWB and the DB parameter to temperature. They explore different proxy and standardization configurations to identify the most suitable approach. They find that results are inconsistent depending on the options tested, but nevertheless can make acceptable reconstructions of past temperature variability. The work is relevant because it tries to increase our knowledge about temperature history by exploiting

proxies that could give us the opportunity to greatly upscale studies phylogenetically, geographically and in terms of replication, in site chronologies.

The major problem with this manuscript is the combination of a great number of models tested and the great variability among the different models tested. Because of this combination it could be argued that the successful models have been achieved spuriously. To attempt to avoid this, I recommend to limit the number of models tested, by performing also climate calibrations with high-pass filtered data (see suggestions below) and to combine this with a discussion of which monthly temperatures can have a causal effect on tree growth. Conducting this additional analysis would narrow down the options that can be tested, but also function as a baseline for the discussion of low-frequency skill in the data. If the high-frequency part of the data is agreeing well with temperature, it is likely safe to assume that a breakdown of agreement when low-frequencies are added is due to low-frequency biases, such as HW-SW-, standardization, etc.. problems. When this is established then tests of how to minimize the loss of signal at lower frequencies can be conducted (different standardizations alternatives). If however, the high-frequency part of the data does not agree very well with temperatures in the first place, it is very unlikely to expect that adding the low-frequency part will contribute with useful information even if correlations are boosted. Therefore, the high-frequency analysis must come first and inform subsequent choices of configurations and options.

A secondary issue is that the authors use reflected BI. This type of data is negatively correlated with what the authors claim to measure in the wood: cell wall, lignin content, but also with the discolorations. If the authors would opt to use the absorbed BI it would let them completely avoid many confused elaborations (see detailed comments below) with regard to standardization and comparisons with MXD etc.

In conclusion, I find the manuscript well written and prepared but I strongly suggest adding a high-frequency analysis, and using absorbed BI. After these revisions and the implementation of the comments below, the manuscript should be suitable for publication.

Detailed comments: L45 Remove "However" L49 replace "for" with "covering" L54-59 This section only discuss the non-climatological variance distorting RW signal, and does not acknowledging that RW and LWB actually may contain different climatic fingerprints. I suggest adding something along this line. L70 Björklund et al worked with absorbed Maximum BI L71-72 Here and in many other places it would be much simpler to start talking about absorbed BI values because these values will be positively correlated with the properties that you mention as potential measurement targets. Why measure the inverted value of what you are interested in? In this way just confuse readers about what you had to do before standardization to make them work properly and why BI is inversely correlated with density etc. L80-81 This is true if we disregard the principle of diminishing records back in time. L92-93 Björklund et al 2014 subtracted average absorption Earlywood BI from maximum absorption BI. L96-98 This sounds like a hypothesis you are going to test later in this paper, but it is not really tested. I would phrase it more like a discussion point: If EWB and LWB contain similar climatic responses and similar standard deviations... L100-101 Not really "another concern". I suggest changes to something like this (I let you worry bout the grammar and English): Finally, although BI based variables hold great promise as an alternative proxy to MXD at inter-annual time-scales, the potential ability of BI to capture decadal to centennial time-scales related to long term-climate changes is still under question. L102 Please clarify if you mean HW-SW color difference L131-134 If you decide to use absorbed BI values this entire section can be removed. If you decide to keep it as is, I strongly recommend to go in to a discussion about why the detrending alternatives are sensitive to this. For example, deterministic detrending such as Neg. exp. or hugershof assume a decline in data values with age. If data values instead have an assumed increase, these methods will be useless. The reason for wanting this added discussion is that some researchers have missed this point and use these methods also for reflected BI. L136-138 I recommend expanding this to also include a more aggressive detrending, perhaps a 25- 35-year spline. This will give more robust climate correlation result. If there are lingering trends in the tree-ring data, and there will be some using

[Figure]

200-year-splines, the risk of spurious trend correlations is relatively high. Adding a high frequency alternative can help to better identify important months for tree-growth. I suspect that some months enter your models just because they have similar trends as the tree-ring data. Also, before performing the aggressive alternative, the climate data should also be detrended similarly. Furthermore, I recommend to restructure the presented results; The high frequency monthly data analysis should be in the main manuscript and the seasonal climate correlations in the supplement together with the low-frequency counterparts. The HW-SW problem will still be present in the analysis using a 200-year spline, if you want to remove this for the analysis you need a softer spline. Rbar, PCA, climate response, between variable correlations should all be done with data with less autocorrelation: softer spline. The low-frequency alternative can be presented on the background of this analysis, but not stand alone. The models' monthly targets for reconstruction should be informed in the first place with high-frequency results. A discussion can be conducted referring to the low-frequency results but not as a major informant of the models. L167 Please specify which function was used to model the regional curve. L171-172 LINres has been shown to create quite some bias in resulting chronologies, see works of Melvin and Briffa, especially if used to model the RC. I instead recommend time-varying response smoother Melvin et al., 2007. L177-181 I suspect the results could be somewhat different with the high-frequency data analysis, see recommendations above. If they are, this is going to be vital information for your main question in the introduction: b) whether meaningful low frequency information can be gleaned from these data? Furthermore, if they are very different, the continuation of the question: "exploiting the long monthly instrumental temperature records that go back into the mid-19 validate secular trends in the TR data" becomes heavily diluted. L198 Again, must be done also with high-frequency data. Should likely cut off some month, and give a better causal reflection of which months are important for radial tree growth. High and persistent correlations with consecutive months makes me suspect trend-correlations. L217-219 Awkward sentence, please rephrase. L227 in both or just in the new one? L265-271 Use absorption BI to avoid confusing comparisons with

MXD.

L272-273 The original DB was introduced in Björklund et al., (2014), but it was further developed in Björklund et al., (2015) were they used a contrast adjustment. More discolored samples had a systematically lower contrast between earlywood and latewood than less discolored samples. If there is a systematic difference in discoloration then this will affect also the traditional DB data. You can easily test if there is a contrast problem in your data with scatterplots of DB vs EWBI, as done in Björklund et al., (2015). If there is a relationship you might at least want to discuss this. If there is not a relationship you will have cleared a question mark.

L276-281 According to my experience the age-trend of MXD would be more similar to DB than LWB. Perhaps different detrending options are needed, but if age-dependent splines are used, as suggested before, these would adapt to the small differences in the data. Neg. exp. or linear functions, for instance, may be directly inappropriate when having juvenile phases of increase and then followed by a decline.

L278 Again use absorption BI.

L283-288 Use absorption BI to avoid having to clarify what you mean.

L288-292 It seems as a contradiction to write that LWB (as temperature proxy) should not have a negative trend w.r.t glacier advancements? The glacier advancement was stable up until 1800 CE and glacier advancement peaked around the turn of the 20th century. Would fit very well with the LWB record that has no trend from 1600-1800 CE and then a negative trend from 1800-1900 CE. The problem would be that there is no pronounced positive trend in the 20th century to melt away the glaciers that expanded prior to this.

L343 Conclusions sections is very long and more like a summary of the discussion

L349-353 I would recommend to test high frequency results before making these bold statements. That is, to first to rule out any trend correlations with winter months for ring

width. After all it is very unusual for ring width to have a broader temperature response than BI or density se e.g. Briffa et al., (2002).
* * *

---

## Referee Comment (RC2) · M. Rydval (Referee) · 9 May 2017

**General comments:**

This manuscript investigates the climatic utility of tree ring data from the Gulf of Alaska. The study experiments with a range of chronology variants utilizing various tree-ring variables (including ring width - RW, latewood blue intensity - LWB and delta blue intensity - DB) and standardization options in an attempt to examine the sensitivity of chronology structure to these choices and to identify the most suitable options for the purpose of developing reconstructions that are minimally biased and accurately represent past climatic variability. The manuscript primarily focuses specifically on the ability of these tree ring variables to accurately represent lower frequency trends. The analysis is performed on tree ring series from a species and a region previously unexplored with respect to the Blue Intensity parameter. In particular, the use of the DB parameter has thus far only been minimally explored in experimental studies. In this sense this work represents an important contribution towards understanding the behaviour of Blue Intensity data and the suitability of such data for reconstructing summer temperatures. On the whole this article is well written, logically structured, without any major fundamental problems and supported by a set of generally clear and relevant figures and tables. Nevertheless, there are a number of specific issues that could be addressed in order to further improve this article and a range of suggestions are detailed below. Some of the limitations of this work include the relatively low series replication for individual sites, limited exploration and discussion of certain aspects of standardisation and chronology development, and some methodological considerations particularly in relation to the BI parameter which could be elaborated on in more detail.

**Specific comments:**

Considering the experimental nature of the LWB and particularly DB parameters, it would have been useful to develop even a limited MXD dataset on at least part of the samples (e.g. from one site) in order to enable a direct comparison of the lower frequency trends in the BI data. Although it is argued that the structure of mountain hemlock wood makes it more difficult to prepare and measure these samples for density, it is not impossible and has been done in past studies. This would have been helpful in evaluating and constraining the utility of differently detrended BI chronologies and therefore considerably benefited this study in further strengthening the case for the use of DB as a better, less biased parameter relative to LWB and a suitable alternative to MXD for this species, especially since this is the first study to measure BI on mountain hemlock samples. Was this option at all considered?

I am somewhat surprised that a higher number of samples was not used for the individual sites. According to Table 1 replication should ideally be 12-28 series for LWB and 14-36 for DB depending on the site. In several cases, the actual maximum number
of series used is below (and in some cases well below) this optimal level. Is this not a problem? The weaker signal strength of BI data and the need for higher replication in order to develop 'robust' chronologies is acknowledged (e.g. L368-369). Also, the relatively low replication may even affect the RW data as stated in L229-230. As stated in the text, a subset of samples was selected for this study from earlier work so why not aim for 25-30 samples per site? That would have at least reduced uncertainty about the representativeness of some of the BI site chronologies, particularly in earlier periods when replication is likely even lower. It would be nice to see a replication plot over time (and EPS plot) for separate sites as well as for the 'all series' pooled version (perhaps as an SI figure) to give a better indication of which periods might potentially be affected by low replication.

L96-98 and L217-219 – What is the rationale for this statement? To my knowledge this issue has not been investigated in any previous study. Presumably a higher correlation between EWB and LWB would imply that EWB expresses in part the same information as LWB, but does that necessarily mean that this information is related to climate? How do you define 'weakly correlated'? Or in other words, what correlation would be acceptable and what would not? Ideally, this statement could be supported with an example and actual data. If nothing else, I would suggest elaborating further on this statement to more clearly express the justification for this claim.

L102-105 As a general comment, some of the limitations of BI (specifically LWB) have already been explored in other studies. Clearly DB is a major improvement, although I wonder just how well DB resolves these issues and specifically whether DB could still have some problems at lower or other frequencies. It is interesting that in some cases the calculation of DB weakens the common signal, suggesting that information which should ideally be preserved is to some extent being removed in the process, yet in other instances the strength of the common signal remains relatively unaffected or even shows improvement. I suppose these questions can only be answered by various future studies that will further explore DB and I would not expect this to be fully
covered here. But perhaps a statement could be included somewhere to caution and emphasize that considerable uncertainty remains with respect to the performance of DB and more work is required in this area.

L109-111 - Is there any indication to what degree early instrumental biases could be a limitation (if at all) in achieving the stated aim?

(L133-138) Is there actually any need to detrend DB series? What is the justification for this? Hypothetically, if both LWB and EWB contain the same ontogenetic trends then by the nature of the DB calculation this trend would be automatically removed. I do not know whether or not that is true. This may be a more complex issue - perhaps only the LWB contains this trend or the LWB and EWB trends related to age differ in some way. But is it not possible that by detrending the DB data some lower frequency climatic information may be unnecessarily removed? Was the development of DB chronologies without performing detrending considered or explored in the analysis? The DB chronology in Figure 5 actually looks like a reasonable chronology variant and so I wonder how non-detrended DB chronologies would perform in terms of calibration and validation statistics relative to the detrended versions.

L146-147 – This is a fairly short validation period. Why not choose an equally long calibration and validation period which has been a common approach in other similar studies? How sensitive are the results to this choice? L244-245 acknowledges that this may be an issue. For example, would a different period affect the significance of any results in Table 4?

L147-148 and top panel in Figure 8 - Why not perform a nested PC reconstruction? By excluding even just one or two of the shortest sites this reconstruction could be extended to the mid or early 18th century, providing more information for comparison with the other reconstructions.

164-173 - Was the variance stabilized to account for changing replication / r-bar through time? Even just a look at the changing variance with replication in Figure S4 in the
supplementary material suggests that this should be performed. Either I have missed this or it is not stated in the methods section.

L171 – Figure S4 suggests that linear detrending may not be the most appropriate choice to detrend for example the DB data. Based on the initial increase in the juvenile period of growth in the DB results, perhaps a somewhat more flexible detrending alternative could be explored to account for this initial increase? Additionally, Figure S4 raises some intriguing questions about non climatic trends in the BI data. I suggest that a fourth panel showing results for EWB should also be shown here. It appears that the initial juvenile trend is present in the EWB only. Does this not suggest that the EWB should not be used to 'correct' LWB in this initial ~30-50 year period?! Because it appears that this initial trend is not present in the LWB data, but is introduced into DB by the EWB data making it necessary to then remove this trend again from the DB series. Due to the experimental nature of this TR variable, several methodological considerations such as the one discussed here remain unaddressed and have not been explored in this study. Considering the nature and aims of this manuscript, I would not expect a detailed evaluation of DB, however, methodological issues such as this and the need to explore them further should at least be acknowledged and more clearly highlighted in the text.

Is it reasonable to develop a RW + DB chronology / reconstruction considering the difference in the seasonal response of RW and BI data and the acknowledgement that RW may not be primarily controlled by summer temperatures in this region (L349-351 and L379-380)?

L187-189 – Maybe already refer to Figure 2 at the end of the first sentence.

Perhaps specifically state somewhere in section 3.3 that JJAS was selected for further analysis for BI (presumably because this is the optimal season).

L196-198 – What might be the reason for such a broad seasonal response in the RW data in this region? Has this been discussed in any of the previous studies (e.g. Wilson
et al. 2007 or Wiles et al. 2014)?

L216 – Just an observation, it is a little surprising that RW agrees more strongly with DB and LWB than DB does with LWB.

L219-220 - Maybe worth clarifying here that EWB would not be expected to contain an actual climate signal in the first place.

L241-242 - Is it possible that the failed validation could be related to the quality of instrumental data for this early period?

L278-280 - This may simply indicate that the use of LWB and EWB to calculate DB is an imperfect procedure. It would be necessary to look at data from other locations (and also other species) to identify whether the DB trends in Figure S4 actually reflect inherent properties related to age (and should therefore be detrended) or are related to other factors. This is an important issue and it is unfortunate that this paper does not or cannot investigate this type of issue in more detail. But perhaps at least a bit more discussion could be included in relation to this?

L294-295 - The results in Figure S4 already indicate that a linear detrending curve can lead to a serious underestimation in the early parts of the series so the poor performance of LINsf is not surprising.

L314-317 - But as discussed in the text (and in relation to the results in lines 286-291) it is apparent that LWB is inherently biased. So why even consider it as a feasible option?

L330 – Maybe include '(Figure 6 and 7)' in the bracket since this reconstruction appears in both figures and is discussed in relation to Figure 7 in the previous paragraph.

L329-331 – It may be worth stating here something along the lines that the 'best performing' PC and extended reconstructions are shown here and compared with Wiles and the glacial advance record - i.e. state the reason why these reconstructions are shown in Figure 8.
L358 – Or for species which do not have this colour difference to begin with.

Figure 3 – It is interesting that LWB calibrates and validates more strongly than DB in terms of the strength of the relationship. Why might this be the case? Could this difference be related to replication?

Figure 6 - It is somewhat difficult to identify the trend of the LINres curve for LWB and especially DB - is it possible to improve the visibility of these curves? Also, in the figure caption (L596-597) maybe consider changing 'low plots' to 'lower set of plots'.

Figure 7 – Why not also show r2 rather than r for the 1850-1900 period?

Table 1 - Is there any real meaning in including N-EPS information for EWB data? Presumably these data do not (or should not) contain any common climatic signal and there would be no point in developing a chronology from these data that would be of much use.

Table 2 – Minor detail, but why not arrange the site order from west to east?

Table 3 – Why is EWB positively (though weakly) correlated with DB?

Technical corrections:

L32 – affecting instead of effecting

L71 – 'as they are a measure of ...' instead of 'as they measure' may be a more accurate statement.

L109 – reconstructions instead of reconstruction

L126 - Please specify the exact calibration target type as there are different versions. IT8.7/1 is a transparency target whereas IT8.7/2 is a reflective target. I assume that the latter was used.

L211-212 – Consider rewording 'potentially optimal' to something along the lines of 'more optimal compared to a PC approach'.

CPD
L218 – Consider specifically stating that the correlation between EWB and LWB is not significant. Also, 'of' missing in 'the utilization DB to'.

L288 – change 'particular' to 'particularly'

L637 - Is there a better word than 'dominant' which could be used here?

---

## Author Comment (AC1) · 1 Jun 2017

WE VERY MUCH APPRECIATE THE DETAILED CRITIQUES FROM BOTH REVIEW-ERS AND WE HOPE WE HAVE ADDRESSED ALL THEIR ISSUES TO BOTH THEIRS AND THE EDITOR'S SATISFACTION. I SHOULD STATE THAT I HAVE ALREADY MADE CHANGES TO THE MANUSCRIPT AND AM HAPPY TO SUBMIT THE EDITED VERSION IF THE EDITOR AGREES WITH THE CHANGES WE HAVE MADE.

Review 1

General comments:

[Figure]

The major problem with this manuscript is the combination of a great number of models tested and the great variability among the different models tested. Because of this combination it could be argued that the successful models have been achieved spuriously. WE THINK THIS IS A LITTLE UNFAIR AS THIS IS AN EXPLORATORY PAPER WHICH PRESENTS A RANGE OF RESULTS (BUILDING FROM SIMPLE RW DATA THROUGH TO A MULTIVARIATE AMALGAM OF USING MULTIPLE TR PARAMETERS) WHICH PROVIDE GUIDANCE FOR THE CONTINUATION OF THIS WORK AS WE EXTEND THESE NEW DATA BACK INTO THE 1ST MILLENNIUM AD. THIS PAPER NEVER AIMED TO DERIVE A NEW DEFINITIVE RECONSTRUCTION FOR THE REGION. THE MAIN CONCLUSION FROM THIS INITIAL LIMITED DATA-SET IS THAT THE DB PARAMETER LIKELY EXPRESSES THE BEST COMPRO-MISE BETWEEN HIGH AND LOW FREQUENCY CALIBRATION FIDELITY (I.E. LWB LOW FREQUENCY IS SEVERELY BIASED). WE FIND IT STRANGE THAT THE RE-VIEWER THINKS THAT WE HAVE ATTAINED A SPURIOUS RESULT AS WE HAVE BEEN CAUTIOUS WITH OUR CONCLUSIONS.

WE ARE HOWEVER WILLING TO MAKE SOME CHANGES TO ALLAY THE RE-VIEWER'S CONCERNS. AN AGE DEPENDENT SPLINE CHRONOLOGY VERSION CAN BE ADDED TO FIGURE 6. FOR FIGURE 7, RATHER THAN USE A SINGLE RE-CONSTRUCTION, WE CAN DERIVE A LWB AND DB BASED GOA COMPOSITE RE-CONSTRUCTION BASED ON ALL THE CHRONOLOGY VARIANTS AS WEIGHTED MEANS RELATED TO THEIR CALIBRATION R2 VALUES TO THE 1901-2010 CAL-IBRATION PERIOD. THE NEW FIGURES 6 AND 7 ARE ATTACHED SEPARATELY. THE RESULTS DO NOT CHANGE WITH THIS APPROACH BUT DOES DISTANCE THE ANALYSIS FROM ANY SUBJECTIVE CHOICES OF USING ONE PARAMETER CHRONOLOGY VARIANT OVER ANOTHER.

To attempt to avoid this, I recommend to limit the number of models tested, by perform-ing also climate calibrations with high-pass filtered data (see suggestions below) and to combine this with a discussion of which monthly temperatures can have a causal effect on tree growth. Conducting this additional analysis would narrow down the options that can be tested, but also function as a baseline for the discussion of low-frequency skill in the data. If the high-frequency part of the data is agreeing well with temperature, it is likely safe to assume that a breakdown of agreement when low-frequencies are added is due to low-frequency biases, such as HW-SW-, standardization, etc.. problems. When this is established then tests of how to minimize the loss of signal at lower frequencies can be conducted (different standardizations alternatives). If however, the high-frequency part of the data does not agree very well with temperatures in the first place, it is very unlikely to expect that adding the low-frequency part will contribute with useful information even if correlations are boosted. Therefore, the high-frequency analysis must come first and inform subsequent choices of configurations and options. DENDROCLIMATOLOGY, UNLIKE MOST OTHER PALAEOCLIMATE APPROACHES HAS THE ABILITY TO DERIVE SO-CALLED ROBUST ESTIMATES OF PAST CLIMATE AT INTER-ANNUAL TO CENTENNIAL TIME-SCALES. ALTHOUGH WE AGREE THAT LOW FREQUENCY TRENDS CAN LEAD TO SPURIOUS CORRELATIONS, WE HAVE BEEN VERY CAREFUL IN TESTING BOTH THE HIGH AND LOW FREQUENCY FIDELITY OF THE MODELS WE PRESENT. THE LOW FREQUENCY FIDELITY IS PARTICULARLY DIFFICULT TO EXAMINE AS WE MUST ASSUME GREATER UNCERTAINTY IN THE EARLY INSTRUMENTAL DATA. IN FACT, THE AMBIGUITY OF THIS PAPER IS THE ASSESSMENT OF THE LOW FREQUENCY SIGNAL IN THE LWB AND DB DATA.

HOWEVER, THE REVIEWER IS ENTIRELY CORRECT THAT WE IDEALLY WANT "EQUAL" COHERENCE WITH PAST TEMPERATURES AT BOTH HIGH AND LOW FREQUENCIES TO THE SAME SEASON. RATHER THAN ADD FURTHER CALIBRATION EXPERIMENTS USING MORE FLEXIBLE DETRENDING OPTIONS, WE FEEL THAT A VALID COMPROMISE IS TO ALSO PRESENT (AS AN EXTRA SUPPLEMENTARY FIGURE) THE CORRELATION RESPONSE FUNCTION ANALYSIS RESULTS OF FIGURE 2 AFTER THE DATA (TR AND TEMPERATURE) HAVE BEEN TRANSFORMED TO 1ST DIFFERENCES. THIS NEW FIGURE VERSION IS ALSO

ATTACHED.

THE RW BASED CORRELATIONS SHOW STRONGEST CORRELATIONS WITH JJA, BUT ARE STILL RELATIVELY HIGH WITH THE BROADER WINDOW OF FEB-AUG. THE REVIEWER IS THEREFORE CORRECT THAT THERE IS SOME SPU-RIOUS TREND RELATED CORRELATION CREEPING IN FOR RW, BUT IS ONLY MINOR AND ARGUABLY IRRELEVANT FOR THIS PAPER AS THE FOCUS OF THE PAPER IS ON THE BLUE INTENSITY BASED PARAMETERS ANYWAY.

THE EWB 1ST DIFFERENCED CORRELATIONS AT INTER-ANNUAL TIME-SCALES ARE NON-SIGNIFICANT. FOR LWB AND DB, THE CORRELATIONS ARE OVERALL SIMILAR AS THE ORIGINAL FIGURE 2 ALTHOUGH THE SEASONS INCLUDING WINTER MONTHS ARE WEAKER. AGAIN – THIS DOES NOT IMPACT THE PA-PER AS THE SUMMER MONTHS WERE THE TARGET CALIBRATION SEASONS. AGAIN, AS WITH THE ORIGINAL FIGURE 2, THE HIGH FREQUENCY CORRELA-TIONS ARE GENERALLY HIGHER FOR LWB THAN DB AND ULTIMATELY. AS IS CLEAR LATER IN THE PAPER, LWB PORTRAY RATHER SPURIOUS LOW FRE-QUENCY TRENDS. THE MAIN ISSUE THEREFORE IS WHICH OF JJA OR JJAS ARE THE OPTIMAL SEASONS TO CALIBRATE AGAINST (JJAS ALWAYS SLIGHTLY BETTER) AND, MORE IMPORTANTLY, HOW ONE COMBINES THE DIFFERENT PA-RAMETERS – I.E. USE RW AND DB IN A MULTIPLE REGRESSION (AS SHOWN HERE), OR UTILISE A BAND-PASS APPROACH AND USE THE RW AND LWB DATA AT THE FREQUENCIES WHERE THEIR SIGNAL IS "ROBUST" – SEE RYDVAL ET AL. 2017 FOR AN EXAMPLE. THIS LATTER APPROACH WILL BE EXPLORED IN A LATER PAPER.

A secondary issue is that the authors use reflected BI. This type of data is negatively correlated with what the authors claim to measure in the wood: cell wall, lignin content, but also with the discolorations. If the authors would opt to use the absorbed BI it would let them completely avoid many confused elaborations (see detailed comments below) with regard to standardization and comparisons with MXD etc. THIS IS A SEMANTIC

COMMENT ABOUT TERMINOLOGY AND DOES NOT IMPACT THE ANALYSIS IN ANY WAY. I BELIEVE THAT OUR METHODOLOGICAL DESCRIPTION IS CLEAR ALTHOUGH A MINOR CLARIFICATION IS POSSIBLE (SEE BELOW).

AS LIGNIN ABSORBS [BLUE] LIGHT, THEN DENSE LATEWOOD (HIGH DENSITY) WILL REFLECT LESS BLUE LIGHT. HENCE RAW LWB AND MXD ARE INVERSELY CORRELATED. THE ONLY REASON THAT RAW LWB NEED TO BE INVERTED IS DUE TO LIMITATIONS OF THE FREELY AVAILABLE DETRENDING SOFTWARE (I.E. ARSTAN) WHERE IT IS THE NORM TO REMOVE NEGATIVE/ZERO SLOPE TRENDS AND RETAIN POSITIVE TRENDS. THEORETICALLY, THE TRENDS IN LWB WILL BE OPPOSITE TO MXD, BUT IN THE SOFTWARE THERE ARE NO OPTIONS TO REMOVE POSITIVE/ZERO SLOPE TRENDS AND RETAIN NEGATIVE ONES. HENCE THE RAW LWB DATA NEED TO BE INVERTED FOR DETRENDING. THIS APPROACH HAS BEEN USED IN WILSON ET AL. (2011, 2014, 2017) AND RY-DVAL ET AL (2014, 2016, 2017) PLUS OTHER PAPERS AND AS FAR AS WE ARE AWARE IT IS ONLY BJÖRKLUND ET AL. WHO HAVE SUGGESTED USING THE TERM "MAXIMUM LATEWOOD BLUE ABSORPTION INTENSITY (MXBI)".

AS A SUBTLE RE-WORDING WE ARE WILLING TO TWEAK THE METHODOLOGI-CAL TEXT AS FOLLOWS:

"RAW EWB AND LWB VARIABLES WERE MEASURED US-ING COORECORDER 8.1 SOFTWARE (CYBIS 2016 - HTTP://WWW.CYBIS.SE/FORFUN/DENDRO/INDEX.HTM), WHICH HAS STATE-OF-THE-ART CAPABILITIES TO ACQUIRE ACCURATE REFLECTANCE INTENSITY RGB COLOUR MEASUREMENTS FROM SCANNED WOOD SAMPLES (SEE RY-DVAL ET AL. 2014). DB VALUES WERE CALCULATED WITHIN COORECORDER BY SUBTRACTING THE RAW LWB VALUES FROM THE RAW EWB VALUES FOR EACH YEAR. SINCE RAW LWB IS NEGATIVELY CORRELATED TO MXD (HIGH DENSITY 'DARK' LATEWOOD = LOW REFLECTANCE), VALUES WERE INVERTED FOLLOWING THE METHOD DETAILED IN RYDVAL ET AL. (2014) TO ALLOW FOR

LWB (HEREAFTER DENOTED AS LWBINV) TO BE DETRENDED IN A SIMILAR WAY TO MXD (SEE ALSO WILSON ET AL. 2014). THE NATURE OF THE DB CALCULATION RESULTS IN THIS PARAMETER BEING POSITIVELY CORRELATED WITH INVERTED LWBINV, SO THESE DATA COULD ALSO BE THEORETICALLY DETRENDED IN A SIMILAR WAY."

AS A FURTHER COMPROMISE TO THE REVIEWER, WE ARE ALSO HAPPY TO ADD IN A CLEAR STATEMENT REFERRING TO BJÖRKLUND ET AL. (2014/2015) STATING THEIR TERMINOLOGY FOR INVERTED LWB.

FINALLY, WE BELIEVE IT IS IMPORTANT TO TREAT BI RELATED PARAMETERS INDEPENDENTLY OF DENSITY. THEY ARE RELATED NO DOUBT, BUT TRYING TO FIT BI TO DENSITY IS A POTENTIALLY DANGEROUS APPROACH. ALTHOUGH MEASURING SIMILAR PROPERTIES, WE CANNOT EXPECT THEM TO BE EXACTLY SIMILAR – ESPECIALLY W.R.T. AGE RELATED TRENDS. ALSO - IT IS NOT REALLY CLEAR WHAT MAXIMUM EARLY WOOD REFLECTANCE (EWB) ACTUALLY REPRESENTS AND IT IS LIKELY THAT THIS PARAMETER IS RELATED TO LUMEN SIZE RATHER THAN ANY PROPERTY REFLECTING SPECIFICALLY COMPOUNDS IN THE EARLYWOOD CELL WALLS.

In conclusion, I find the manuscript well written and prepared but I strongly suggest adding a high-frequency analysis, and using absorbed BI. After these revisions and the implementation of the comments below, the manuscript should be suitable for publication. We hope the 1st differenced based results shown above can address the reviewer's concerns on the FIRST POINT AND WE FEEL NO OBLIGATION TO CHANGE OUR TERMINOLOGY TO ADDRESS THE SECOND POINT AS IT DOES NOT CHANGE THE ANALYSIS/RESULTS IN ANY WAY AND WE FEEL OUR CURRENT DESCRIPTION IS ADEQUATE AND CONSISTENT WITH MOST PREVIOUS PUBLICATIONS.

Detailed comments:

L45 Remove "However" AGREED – THIS CAN BE DONE WHEN "ALLOWED" TO EDIT THE PAPER FOR FINAL PUBLICATION

L49 replace "for" with "covering" AGREED – THIS CAN BE DONE WHEN "ALLOWED" TO EDIT THE PAPER FOR FINAL PUBLICATION

L54-59 This section only discuss the non-climatological variance distorting RW signal, and does not acknowledging that RW and LWB actually may contain different climatic fingerprints. I suggest adding something along this line. PLEASE SEE ATTACHED 1ST DIFFERENCED CORRELATION RESPONSE FUNCTION ANALYSIS RESULTS. THESE CAN BE ADDED TO THE SUPPLEMENTARY AND APPROPRIATE DISCUSSION ADDED.

L70 Björklund et al worked with absorbed Maximum BI BJÖRKLUND'S ABSORBED MAXIMUM BI IS THE SAME AS THE INVERTED LWB USED IN THIS PAPER. WE FEEL THE CURRENT DESCRIPTION IS CLEAR ENOUGH – ESP. WITH SUGGESTIONS MENTIONED ABOVE.

L71-72 Here and in many other places it would be much simpler to start talking about absorbed BI values because these values will be positively correlated with the properties that you mention as potential measurement targets. Why measure the inverted value of what you are interested in? In this way just confuse readers about what you had to do before standardization to make them work properly and why BI is inversely correlated with density etc. SEE COMMENTS ABOVE. I AM SURE BJÖRKLUND MEASURED RAW INTENSITY VALUES OF BLUE REFLECTANCE AND THEN INVERTED THE DATA AS WE HAVE DONE. THE DIFFERENCE AFTER THAT IS SEMANTIC ONLY.

L80-81 This is true if we disregard the principle of diminishing records back in time. THIS HOLDS TRUE DESPITE THE REDUCTION IN REPLICATION BACK IN TIME. THE COMMUNITY NEEDS TO BE CAREFUL AS TO DEFINE CLEAR THRESHOLD OF TRUNCATION. IN THIS PAPER, THE DATA ARE FAR FROM IDEAL W.R.T. REPLI-

CATION – THAT WAS PARTLY STRATEGIC. THE FACT THAT THE RESULTS ARE ENCOURAGING SUGGESTS THAT CALIBRATION/VERIFICATION WILL IMPROVE SUBSTANTIALLY AS REPLICATION IS INCREASED. NOT SURE THIS COMMENT WARRANTS ANY SPECIFIC CHANGE.

L92-93 Björklund et al 2014 subtracted average absorption Earlywood BI from maximum absorption BI. THIS AGAIN HIGHLIGHTS WHY WE PREFER OUR CURRENT METHODOLOGICAL DESCRIPTION. WE HAVE USED THE RAW EWB AND LWB VALUES AND USED THE DIFFERENCE TO DERIVE THE DB VALUE. THIS IS MATHEMATICALLY THE SAME AS WHAT THE REVIEWER WANTS US TO IMPLEMENT, BUT WE SEE NO GAIN WITH SUCH A CHANGE AS WE ARE DOING THE SAME METHOD, BUT USING DIFFERENT TERMINOLOGY WHICH IS CLEARLY DEFINED.

L96-98 This sounds like a hypothesis you are going to test later in this paper, but it is not really tested. I would phrase it more like a discussion point: If EWB and LWB contain similar climatic responses and similar standard deviations. . . AGREED – THIS IS A LITTLE VAGUE AND REVIEWER 2 ALSO FLAGGED THIS. WE WOULD GLADLY CHANGE THIS TEXT (AND ASSOCIATED LATER DISCUSSION) TO BASICALLY HIGHLIGHT THAT IF EWB AND LWB BOTH EXPRESS THE SAME CLIMATIC RESPONSE (THIS SHOULD NOT BE THE CASE), THE RESULTANT DERIVED DB DATA WILL NOT SHOW THIS COMMON RESPONSE AND LIKELY BE INFERIOR TO LWB. THIS IS AN EVOLVING PROPERTY OF DB AND CERTAINLY NEEDS MORE EXAMINATION USING MORE SPECIES AND LOCATIONS.

L100-101 Not really "another concern". I suggest changes to something like this (I let you worry bout the grammar and English): Finally, although BI based variables hold great promise as an alternative proxy to MXD at inter-annual time-scales, the potential ability of BI to capture decadal to centennial time-scales related to long term-climate changes is still under question. AGREED – HAPPY TO RE-WRITE THIS SECTION AND CLARIFY THE MESSAGE BETTER.

L102 Please clarify if you mean HW-SW color difference AGREED. YES – HW-SW – THIS CAN BE CLARIFIED WHEN "ALLOWED" TO EDIT THE PAPER FOR FINAL PUBLICATION

L131-134 If you decide to use absorbed BI values this entire section can be removed. If you decide to keep it as is, I strongly recommend to go in to a discussion about why the detrending alternatives are sensitive to this. For example, deterministic detrending such as Neg. exp. or hugershof assume a decline in data values with age. If data values instead have an assumed increase, these methods will be useless. The reason for wanting this added discussion is that some researchers have missed this point and use these methods also for reflected BI. AS DISCUSSED ABOVE, WE ARE HAPPY WITH OUR CURRENT TERMINOLOGY AND METHODS DESCRIPTION.

L136-138 I recommend expanding this to also include a more aggressive detrending, perhaps a 25- 35-year spline. This will give more robust climate correlation result. If there are lingering trends in the tree-ring data, and there will be some using 200-year-splines, the risk of spurious trend correlations is relatively high. Adding a high frequency alternative can help to better identify important months for tree-growth. I suspect that some months enter your models just because they have similar trends as the tree-ring data. Also, before performing the aggressive alternative, the climate data should also be detrended similarly. Furthermore, I recommend to restructure the presented results; The high frequency monthly data analysis should be in the main manuscript and the seasonal climate correlations in the supplement together with the low-frequency counterparts. The HW-SW problem will still be present in the analysis using a 200-year spline, if you want to remove this for the analysis you need a softer spline. Rbar, PCA, climate response, between variable correlations should all be done with data with less autocorrelation: softer spline. The low-frequency alternative can be presented on the background of this analysis, but not stand alone. The models' monthly targets for reconstruction should be informed in the first place with high-frequency results. A discussion can be conducted referring to the low-frequency results but not as

a major informant of the models. THE MEAN/MEDIAN SEGMENT LENGTHS OF ALL SITES IS > 200 YEARS SO ANY "LINGERING" TRENDS FOR INDIVIDUAL SERIES WOULD BE MINIMAL USING A 200-YEAR SPLINE. THAT IS WHY WE CHOSE IT. WE CAN ADD A COMMENT TO THIS END.

WE HOPE THE ADDED 1ST DIFFERENCED BASED CRFA WILL ADDRESS THE REVIEWER'S CONCERNS AS TO OUR ANALYSIS AND THE IDENTIFICATION OF THE "CORRECT" MONTH FOR CALIBRATION.

THE REVIEWER IS SUGGESTING A MAJOR RE-ANALYSIS HERE AND IT IS NOT CLEAR WHAT GAIN THERE WOULD BE. THE 200-YR SPLINE APPROACH IS A COMPROMISE BETWEEN MINIMISING THE HW-SW BIAS WHILE RETAINING SOME REALISTIC MULTI-DECADAL INFORMATION. WE SEE NO JUSTIFICATION OF RE-DOING THE ANALYSIS USING A MUCH MORE FLEXIBLE SPLINE.

L167 Please specify which function was used to model the regional curve. YES – SORRY – WE CAN ADD THIS INFORMATION IN. THE REGIONAL CURVE WAS SMOOTHED WITH A SPLINE OF 10% THE RC SERIES LENGTH AND THAT FUNC-TION USED FOR DETRENDING.

L171-172 LINres has been shown to create quite some bias in resulting chronologies, see works of Melvin and Briffa, especially if used to model the RC. I instead recom-mend time-varying response smoother Melvin et al., 2007. AGREED THAT A LINRES APPROACH MAY IMPART BIASES, BUT WE WOULD ARGUE THAT ALL DETREND-ING APPROACHES HAVE THEIR OWN BIASES. THAT IS WHY WE HAVE PRE-SENTED A SMALL SUB-SET OF POSSIBLE DETRENDING CHOICES. WE COULD HAVE EXPANDED ON THIS SUBSTANTIALLY, BUT THAT WILL BE A FOCUS WHEN WE HAVE THE FULL 1000+ YEAR DATA-SET. I WOULD LIKE TO REMIND THE RE-VIEWER THAT THE LINRES APPROACH HAS BEEN USED ON MANY DENSITY BASED STUDIES IN THE PAST.

MORE IMPORTANTLY, THE AGE-DEPENDENT SPLINE APPROACH OF MELVIN ET

[Figure]

AL. (2007) IS VERY MUCH AN UNTESTED DETRENDING APPROACH. I HAVE EX-
PERIMENTED WITH THIS OPTION AND (1) IT CAN OFTEN INFLATE RECENT PE-
RIOD VALUES AND (2) IMPLICITLY WILL REMOVE SECULAR SCALE VARIATION –
IT IS STILL A SPLINE APPROACH.

HOWEVER, WE ARE HAPPY TO ADD IN AN AGE DEPENDENT SPLINE VERSION
INTO THE MIX FOR FIGURES 6 AND 7 – SEE ATTACHED FIGURES. TABLE 5
CAN BE UPDATED – SEE ALSO ATTACHED FIGURE. THE R2 VALUES FOR THE
1901-2010 PERIOD CAN THEN BE USED AS WEIGHTING TERM TO COMBINE ALL
THESE DIFFERENT VARIANTS TO DERIVE GOA WEIGHTED COMPOSITES FOR
PARAMETER. THE UPDATED FIGURE 7 IS ATTACHED. THE UPDATED FIGURE 8
(WHICH NOW INCLUDES THE WEIGHTED LWB DATA) IS ALSO ATTACHED, WHICH
CLEARLY SHOWS THAT DESPITE REASONABLE CALIBRATION AND VERIFICA-
TION (FIGURE 7, TABLE 5), THE LOW FREQUENCY TRENDS OF THE LWB DATA
DO APPEAR AT ODDS WITH THE OTHER DATA-SETS. WE WILL EXPAND THE DIS-
CUSSION ON OTHER STUDIES INDICATING A COOLER RATHER THAN WARMER
LITTLE ICE AGE.

L177-181 I suspect the results could be somewhat different with the high-frequency
data analysis, see recommendations above. If they are, this is going to be vital infor-
mation for your main question in the introduction: b) whether meaningful low frequency
information can be gleaned from these data? Furthermore, if they are very different,
the continuation of the question: "exploiting the long monthly instrumental temperature
records that go back into the mid-19 validate secular trends in the TR data" becomes
heavily diluted. THE CURRENT ANALYSIS, AS A FIRST ATTEMPT, ALREADY AD-
DRESSES THE LOW FREQUENCY ISSUE WITH APPROPRIATE DISCUSSION AND
THE 1ST DIFFERENCE RESULTS VALIDATE WELL THAT THE APPROPRIATE SEA-
SON HAS BEEN TARGETED.

ULTIMATELY, IT WILL NEVER BE POSSIBLE TO IDENTIFY THE "CORRECT" DE-
TRENDING APPROACH AND WHEN THE FINAL DATA HAVE BEEN FINALISED

WE WILL APPROACH THE RECONSTRUCTION USING A SIMILAR ENSEMBLE AP-PROACH AS INTRODUCED BY WILSON ET AL. (2014) FOR A RELATED STUDY IN THE CANADIAN ROCKIES. THIS WILL ALLOW DETRENDING UNCERTAINTY TO BE EVALUATED IN THE ERROR ESTIMATES. AT THIS TIME, WE JUST WANTED TO HIGHLIGHT THE SENSITIVITY TO SUCH METHODOLOGICAL CHOICES – FIG-URE 6 DOES THIS WELL.

L198 Again, must be done also with high-frequency data. Should likely cut off some month, and give a better causal reflection of which months are important for radial tree growth. High and persistent correlations with consecutive months makes me suspect trend-correlations. SEE ABOVE. THE 1ST DIFFERENCED CRFA ADDRESSES THIS.

L217-219 Awkward sentence, please rephrase. THIS SENTENCE CAN BE TWEAKED WHEN "ALLOWED" TO EDIT THE PAPER FOR FINAL PUBLICATION

L227 in both or just in the new one? WE BELIEVE THE CURRENT WORDING IS QUITE CLEAR THAT THIS IS ONLY FOR THE LATTER "NEW DATA" RW BASED RECON.

L265-271 Use absorption BI to avoid confusing comparisons with MXD. SEE COM-MENTS ABOVE.

L272-273 The original DB was introduced in Björklund et al., (2014), but it was fur-ther developed in Björklund et al., (2015) were they used a contrast adjustment. More discolored samples had a systematically lower contrast between earlywood and late-wood than less discolored samples. If there is a systematic difference in discoloration then this will affect also the traditional DB data. You can easily test if there is a con-trast problem in your data with scatterplots of DB vs EWBI, as done in Björklund et al., (2015). If there is a relationship you might at least want to discuss this. If there is not a relationship you will have cleared a question mark. THIS IS A GOOD POINT, BUT IS NOT RELEVANT TO THE ANALYSIS PERFORMED FOR THIS CURRENT PAPER. WE BELIEVE, FOR THE CURRENT DATA ADAPTIVE DETRENDING TECH-

NIQUES USED HEREIN, THAT THIS ISSUE IS NOT YET RELEVANT, BUT, WILL BECOME RELEVANT AS WE INCORPORATE SNAG AND SUB-FOSSIL MATERIAL TO EXTEND THE REGIONAL COMPOSITE CHRONOLOGY BACK IN TIME.

L276-281 According to my experience the age-trend of MXD would be more similar to DB than LWB. Perhaps different detrending options are needed, but if age-dependent splines are used, as suggested before, these would adapt to the small differences in the data. Neg. exp. or linear functions, for instance, may be directly inappropriate when having juvenile phases of increase and then followed by a decline. AS PRE-VIOUSLY MENTIONED THE CURRENT "CONSERVATIVE" APPROACHES AIM TO RETAIN LOW FREQUENCY INFORMATION. WE BELIEVE THE AGE DEPENDENT SPLINE WILL REMOVE SUCH TRENDS. HOWEVER, AN AGE DEPENDENT SPLINE OPTION IS NOW INCLUDED IN THE ANALYSIS. IT DOES NOT CHANGE ANY OF THE RESULTS!

L278 Again use absorption BI. SEE ABOVE.

L283-288 Use absorption BI to avoid having to clarify what you mean. NO – SEE ABOVE. THE DISCUSSION WILL NOT CHANGE BY USING DIFFERENT TERMI-NOLOGY.

L288-292 It seems as a contradiction to write that LWB (as temperature proxy) should not have a negative trend w.r.t glacier advancements? The glacier advancement was stable up until 1800 CE and glacier advancement peaked around the turn of the 20th century. Would fit very well with the LWB record that has no trend from 1600-1800 CE and then a negative trend from 1800-1900 CE. The problem would be that there is no pronounced positive trend in the 20th century to melt away the glaciers that expanded prior to this. WE BELIEVE THE REVIEWER IS PERHAPS A LITTLE CONFUSED HERE. ALL LWB VARIANTS IMPLY WARMER THAN AVERAGE (20TH CENTURY) TEMPERATURES PRIOR TO 1850. THIS DOES NOT FIT WITH THE GLACIAL EX-PANSION DATA SHOWN IN FIGURE 8. THE DB BASED RECONSTRUCTION (AND

VARIANTS) HOWEVER, DENOTE COOLER THAN AVERAGE (20TH CENTURY) TEMPERATURES PRIOR TO 1850 WHICH IS IN LINE WITH COOLER CONDITIONS NEEDED FOR CONTINUED GLACIAL EXPANSION THROUGH THIS PERIOD.

L343 Conclusions sections is very long and more like a summary of the discussion 1.5 PAGES OF SUMMARY AND RECOMMENDATIONS APPEAR APPROPRIATE TO US. IN FACT, A LITTLE MORE DISCUSSION MAY BE ADDED ABOUT POTENTIAL FUTURE STRATEGIES TO ADDRESS THE HIGH AND LOW FREQUENCY ISSUES OF THE LWB VS DB VS RW DATA.

L349-353 I would recommend to test high frequency results before making these bold statements. That is, to first to rule out any trend correlations with winter months for ring-width. After all it is very unusual for ring width to have a broader temperature response than BI or density se e.g. Briffa et al., (2002). BEYOND THE SCOPE OF THIS PAPER, IF THE RW DATA ARE REGRESSED ON THE DB DATA, THE RESIDUALS FROM THIS ANALYSIS CORRELATE WITH WINTER TEMPERATURES. SEE WILSON ET AL. (2007) WHERE THE RW COMPOSITE CLEARLY PICKS UP DECADE SCALE SHIFTS SEEN IN THE PDO AND OTHER METRICS OF PACIFIC DECADAL VARIABILITY. SUCH SHIFTS ARE NOT SEEN IN THE BI BASED PARAMETERS.

[Figure]

[Figure]

**Fig. 1.**

Weighted LWB$_{inv}$

r = 0.52
RE = 0.49
CE = 0.16

$r^2$ = 0.43
DW = 1.42*
Lin r = 0.30$^{\#}$

Weighted DB

r = 0.47
RE = 0.46
CE = 0.11

$r^2$ = 0.40
DW = 1.54
Lin r = 0.39$^{\#}$

JJAS temperature anomalies (w.r.t. 1961–1990)

Calendar Years

**Fig. 2.**

[Figure]

**Fig. 3.**

[Figure]

|      | 1901-2010 Calibration | | | | | 1850-1900 Validation | | |
| --- | --- | --- | --- | --- | --- | --- | --- | --- |
|      | series entered | r | r2 | DW | LINr | r | RE | CE |
| LWB | LINres | 0.64 | 0.41 | 1.36 | 0.36 | 0.53 | 0.44 | 0.07 |
|      | RCSres | 0.26 | 0.07 | 1.28 | 0.48 | 0.56 | 0.01 | -0.64 |
|      | LINsf | 0.64 | 0.41 | 1.37 | 0.36 | 0.53 | 0.43 | 0.06 |
|      | RCSsf | 0.21 | 0.05 | 1.32 | 0.46 | 0.56 | -0.05 | -0.73 |
|      | ADSsf | 0.69 | 0.47 | 1.58 | 0.06 | 0.50 | 0.51 | 0.20 |

|      | 1901-2010 Calibration | | | | | 1850-1900 Validation | | |
| --- | --- | --- | --- | --- | --- | --- | --- | --- |
|      | series entered | r | r2 | DW | LINr | r | RE | CE |
| DB | LINres | 0.55 | 0.31 | 1.37 | 0.50 | 0.50 | 0.52 | 0.21 |
|      | RCSres | 0.64 | 0.40 | 1.59 | 0.40 | 0.48 | 0.50 | 0.18 |
|      | LINSF | 0.54 | 0.29 | 1.35 | 0.38 | 0.43 | 0.40 | 0.00 |
|      | RCSsf | 0.65 | 0.43 | 1.64 | 0.35 | 0.47 | 0.48 | 0.15 |
|      | ADSsf | 0.65 | 0.42 | 1.59 | 0.30 | 0.47 | 0.34 | -0.09 |

**Fig. 4.**

[Figure]

**Fig. 5.**

---

## Author Comment (AC2) · 1 Jun 2017

WE VERY MUCH APPRECIATE THE DETAILED CRITIQUES FROM BOTH REVIEWERS AND WE HOPE WE HAVE ADDRESSED ALL THEIR ISSUES TO BOTH THEIRS AND THE EDITOR'S SATISFACTION. I SHOULD STATE THAT I HAVE ALREADY MADE CHANGES TO THE MANUSCRIPT AND AM HAPPY TO SUBMIT THE EDITED VERSION IF THE EDITOR AGREES WITH THE CHANGES WE HAVE MADE.

Review 2

Specific comments:

[Figure]

Considering the experimental nature of the LWB and particularly DB parameters, it would have been useful to develop even a limited MXD dataset on at least part of the samples (e.g. from one site) in order to enable a direct comparison of the lower frequency trends in the BI data. Although it is argued that the structure of mountain hemlock wood makes it more difficult to prepare and measure these samples for density, it is not impossible and has been done in past studies. This would have been helpful in evaluating and constraining the utility of differently detrended BI chronologies and therefore considerably benefited this study in further strengthening the case for the use of DB as a better, less biased parameter relative to LWB and a suitable alternative to MXD for this species, especially since this is the first study to measure BI on mountain hemlock samples. Was this option at all considered? MXD HAS NOT BEEN MEASURED ON MOUNTAIN HEMLOCK TREES IN THIS REGION AS FAR AS WE ARE AWARE. ALTHOUGH WE AGREE THEORETICALLY WITH THE RE-VIEWER'S COMMENT THAT THIS COMPARISON COULD BE USEFUL, MXD WAS NEVER FACTORED INTO THE RESEARCH DESIGN OR BUDGET AND THE FO-CUS OF THE STUDY WAS ALWAYS ON BI AND RELATED PARAMETERS. WE (AS A COMMUNITY) SHOULD ALSO BE CAREFUL NOT TO ASSUME THAT MXD IS THE "TRUTH" SO HAVE ASSESSED THE DATA AS BEST AS POSSIBLE USING INSTRUMENTAL DATA AND OTHER PROXY ARCHIVES FOR THE REGION.

I am somewhat surprised that a higher number of samples was not used for the individual sites. According to Table 1 replication should ideally be 12-28 series for LWB and 14-36 for DB depending on the site. In several cases, the actual maximum number of series used is below (and in some cases well below) this optimal level. Is this not a problem? The weaker signal strength of BI data and the need for higher replication in order to develop 'robust' chronologies is acknowledged (e.g. L368-369). Also, the relatively low replication may even affect the RW data as stated in L229-230. As stated in the text, a subset of samples was selected for this study from earlier work so why not aim for 25-30 samples per site? That would have at least reduced uncertainty about the representativeness of some of the BI site chronologies, particularly in earlier periods when replication is likely even lower. It would be nice to see a replication plot over time (and EPS plot) for separate sites as well as for the 'all series' pooled version (perhaps as an SI figure) to give a better indication of which periods might potentially be affected by low replication. THIS IS AN EXPLORATIVE PAPER AND WE HAVE BEEN CLEAR, AS THE REVIEWER STATES, THAT THE NUMBER OF RECORDS ARE LOWER THAN WOULD BE IDEAL. DATA DEVELOPMENT IS ON-GOING (DATA BEING ADDED CONTINUALLY). HOWEVER, WE WANTED TO WRITE THIS INITIAL EXPLORATIVE PAPER TO PARTLY INFORM OURSELVES AS TO THE CONTINUED STRATEGY OF THIS WORK AS WELL AS THE WIDER COMMUNITY. PREVIOUS RW BASED CALIBRATIONS (WILSON ET AL 2007; WILES ET AL. 2014) SHOW THAT WHEN REPLICATION IS HIGH, WE CAN EXPLAIN AROUND 40% OF THE TEMPERATURE VARIANCE. USING THIS NON-IDEAL DATA-SET WE ONLY EXPLAIN 27% FOR RW, BUT >40% USING LWB OR DB. THIS SUGGESTS THAT THE RESULTS CAN ONLY IMPROVE AS WE INCREASE REPLICATION. WE ARE HAPPY TO MAKE A STRONGER SPECIFIC COMMENT IN THIS REGARD.

L96-98 and L217-219 – What is the rationale for this statement? To my knowledge this issue has not been investigated in any previous study. Presumably a higher correlation between EWB and LWB would imply that EWB expresses in part the same information as LWB, but does that necessarily mean that this information is related to climate? How do you define 'weakly correlated'? Or in other words, what correlation would be acceptable and what would not? Ideally, this statement could be supported with an example and actual data. If nothing else, I would suggest elaborating further on this statement to more clearly express the justification for this claim. THIS WAS BROUGHT UP BY REVIEW 1 AND WE AGREE THE CURRENT WORDING IS AMBIGUOUS AND NEEDS CLARIFICATION. BASICALLY WE WANT TO PROVIDE A THEORETICAL BASIS THAT DB WILL LIKELY BE USEFUL WHEN EWB AND LWB ARE UNCORRELATED AT HIGH FREQUENCIES.

L102-105 As a general comment, some of the limitations of BI (specifically LWB) have

already been explored in other studies. Clearly DB is a major improvement, although I wonder just how well DB resolves these issues and specifically whether DB could still have some problems at lower or other frequencies. It is interesting that in some cases the calculation of DB weakens the common signal, suggesting that information which should ideally be preserved is to some extent being removed in the process, yet in other instances the strength of the common signal remains relatively unaffected or even shows improvement. I suppose these questions can only be answered by various future studies that will further explore DB and I would not expect this to be fully covered here. But perhaps a statement could be included somewhere to caution and emphasize that considerable uncertainty remains with respect to the performance of DB and more work is required in this area. THIS IS A VALID COMMENT AND WE SHOULD EMPHASISE BETTER THAT IN FACT THE YEAR-TO-YEAR SIGNAL IN THE LWB DATA IS OFTEN STRONGER THAN THE DB DATA. ALTHOUGH NOT RELEVANT FOR THIS SPECIFIC PAPER, A SIMILAR BAND-PASS APPROACH AS UTILISED BY RYDVAL ET AL. (2016 AND 2017) COULD WORK VERY WELL IN THE GOA REGION. WE CAN ADD MORE DISCUSSION IN THIS REGARD BUT THIS WILL BE THE FOCUS FOR A LATER PAPER.

L109-111 - Is there any indication to what degree early instrumental biases could be a limitation (if at all) in achieving the stated aim? THIS IS AN IMPORTANT POINT AND WE ARE HAPPY TO ADD FURTHER DISCUSSION ABOUT THE ISSUES OF USING EARLY INSTRUMENTAL DATA. THE MAJOR ISSUE HERE IS THE LARGE DIFFERENCE BETWEEN LWB (WARM LIA) AND DB (COOL LIA) WHICH CANNOT BE ADDRESSED USING THE LONGER INSTRUMENTAL DATA. WE MUST RELY ON OTHER PROXY RECORDS FROM THE REGION (I.E. GLACIAL RECORD) WHICH SUGGEST A LIA ABOUT 1 DEGREE COOLER THAN THE 20TH CENTURY WHICH FIT WELL WITH THE DB AND RW DATA.

(L133-138) Is there actually any need to detrend DB series? What is the justification for this? Hypothetically, if both LWB and EWB contain the same ontogenetic trends

then by the nature of the DB calculation this trend would be automatically removed. I do not know whether or not that is true. This may be a more complex issue - perhaps only the LWB contains this trend or the LWB and EWB trends related to age differ in some way. But is it not possible that by detrending the DB data some lower frequency climatic information may be unnecessarily removed? Was the development of DB chronologies without performing detrending considered or explored in the analysis? The DB chronology in Figure 5 actually looks like a reasonable chronology variant and so I wonder how non-detrended DB chronologies would perform in terms of calibration and validation statistics relative to the detrended versions. THIS IS A VALID POINT. THEORETICALLY, ONE WOULD ASSUME THAT DB WOULD BE SIMILAR TO MXD IN TREND, BUT I DON'T THINK THE EXPERIMENTATION WITH BI BASED PARAMETERS ARE ADVANCED ENOUGH TO ADDRESS THIS. I DON'T THINK WE CAN ASSUME THAT EWB AND LWB NECESSARILY WILL SHOW THE SAME ONTOGENETIC TREND. APPENDIX FIGURE 4 CLEARLY SHOWS A MORE COMPLEX MEAN AGE RELATED CURVE FOR DB THAN LWB, BUT THE LATTER WILL BE IMPACTED BY THE HW-SW COLOUR CHANGES, SO IT IS DIFFICULT TO JUDGE THIS SPECIFICALLY. TO PARTLY ASSESS THIS, AN AGE DEPENDENT SPLINE IS NOW ALSO USED IN THE ANALYSIS PRESENTED IN FIGURE 6 AND 7. SEE COMMENTS FOR REVIEWER 1.

L146-147 – This is a fairly short validation period. Why not choose an equally long calibration and validation period which has been a common approach in other similar studies? How sensitive are the results to this choice? THE JUSTIFICATION FOR THESE PERIODS WAS NOT MADE AND THIS SHOULD BE CLARIFIED. AS MUCH OF THE TR DATA IN ALASKA EXPERIENCE CALIBRATION AND VALIDATION ISSUES, SPECIFICALLY IN THE RECENT PERIOD (SO CALLED DIVERGENCE), THIS APPROACH WAS USED FOR THE INITIAL CALIBRATION/VALIDATION TESTS. THERE WILL CERTAINLY BE SOME SENSITIVITY TO THE PERIODS USED, BUT IT SHOULD NOT BE FORGOTTEN THAT THE FULL PERIOD CALIBRATION (1901-2010) WAS VALIDATED AGAINST THE 19TH CENTURY DATA (FIGURE 7), SO MUL-

TIPLE PERIODS HAVE BEEN USED.

L244-245 acknowledges that this may be an issue. For example, would a different period affect the significance of any results in Table 4? WE BELIEVE THAT CONSISTENCY OF THE METHOD FOR TESTING EACH OF THE PARAMETERS AND THEIR COMBINATIONS IS MORE IMPORTANT THAN USING DIFFERENT PERIODS PER SE. THIS IS A VALID POINT HOWEVER AND CERTAINLY THE RESULTS WILL CHANGE SOMEWHAT IF DIFFERENT PERIODS WERE USED. HOWEVER, THIS WAS NOT THE AIM OF THE PAPER AND THE RESULTS OF TABLE 4 AND 5 MUST BE POOLED TOGETHER TO DERIVE AN OBJECTIVE ASSESSMENT OF THE VARYING STRENGTHS AND WEAKNESSES OF THESE DIFFERENT PARAMETERS.

L147-148 and top panel in Figure 8 - Why not perform a nested PC reconstruction? By excluding even just one or two of the shortest sites this reconstruction could be extended to the mid or early 18th century, providing more information for comparison with the other reconstructions. IT IS NOT THE SPECIFIC AIM, AT THIS TIME, TO DERIVE A NEW DEFINITIVE RECONSTRUCTION SPECIFICALLY. THIS PAPER DETAILS MULTIPLE EXPERIMENTS TO EXPLORE THE UTILITY OF THESE DIFFERENT PARAMETERS FOR CLIMATE RECONSTRUCTION AND TO HIGHLIGHT POTENTIAL PARAMETER SPECIFIC BIASES. IT IS NOT CLEAR WHY A NESTING APPROACH WOULD CHANGE THE CONCLUSIONS ALREADY DISCUSSED IN THE PAPER.

164-173 - Was the variance stabilized to account for changing replication / r-bar through time? Even just a look at the changing variance with replication in Figure S4 in the supplementary material suggests that this should be performed. Either I have missed this or it is not stated in the methods section. YES – SORRY – VARIANCE STABILISATION WAS PERFORMED AND THIS SHOULD BE MENTIONED IN THE METHODOLOGY.

L171 – Figure S4 suggests that linear detrending may not be the most appropriate

choice to detrend for example the DB data. Based on the initial increase in the juvenile period of growth in the DB results, perhaps a somewhat more flexible detrending alternative could be explored to account for this initial increase? Additionally, Figure S4 raises some intriguing questions about non climatic trends in the BI data. I suggest that a fourth panel showing results for EWB should also be shown here. It appears that the initial juvenile trend is present in the EWB only. Does this not suggest that the EWB should not be used to 'correct' LWB in this initial âĹij30-50 year period?! Because it appears that this initial trend is not present in the LWB data, but is introduced into DB by the EWB data making it necessary to then remove this trend again from the DB series. Due to the experimental nature of this TR variable, several methodological considerations such as the one discussed here remain unaddressed and have not been explored in this study. Considering the nature and aims of this manuscript, I would not expect a detailed evaluation of DB, however, methodological issues such as this and the need to explore them further should at least be acknowledged and more clearly highlighted in the text. ATTACHED AN EXTENDED FIGURE S4 WHICH NOW INCLUDES THE EWB AGE ALIGNED CURVE. THE REVIEWER IS CORRECT THAT THERE REMAINS MUCH AMBIGUITY W.R.T. TRENDS AS A FUNCTION OF AGE. WE AGREE THAT LINEAR FUNCTIONS MAY NOT REPRESENT THE BEST "FIT" FOR DETRENDING AND REVIEWER 1 MENTIONED THIS ISSUE AS WELL. WE USED THE CURRENT CHOICES AS THEY ARE THE COMMON APPROACH IN THE LITERATURE. HOWEVER, WE HAVE NOW ADDED AN AGE DEPENDENT SPLINE OPTION INTO THE MIX. SEE OTHER UPDATED FIGURES W.R.T. REVIEWER 1.

Is it reasonable to develop a RW + DB chronology / reconstruction considering the difference in the seasonal response of RW and BI data and the acknowledgement that RW may not be primarily controlled by summer temperatures in this region (L349-351 and L379-380)? PLEASE SEE THE RESULTS FROM THE 1ST DIFFERENCED CORRELATION RESPONSE FUNCTION ANALYSIS RESULTS. YES – WE BELIEVE THAT THESE PARAMETERS CAN BE COMBINED ALTHOUGH PLEASE NOTE THAT THE RECONSTRUCTION PRESENTED IN FIGURE 7 ARE SINGLE PARAMETER

ONLY.

L187-189 – Maybe already refer to Figure 2 at the end of the first sentence. THIS SECTION FOCUSSES ONLY ON THE COMMON SIGNAL AND NOT THE CLIMATE SIGNAL EXPRESSED BY IT. FIGURE 2 IS RELEVANT FOR THE NEXT SECTION.

Perhaps specifically state somewhere in section 3.3 that JJAS was selected for further analysis for BI (presumably because this is the optimal season). YES – THIS CAN BE DONE.

L196-198 – What might be the reason for such a broad seasonal response in the RW data in this region? Has this been discussed in any of the previous studies (e.g. Wilson et al. 2007 or Wiles et al. 2014)? YES – THIS WAS DISCUSSED IN PREVIOUS PAPERS AND IS NOT RELEVANT TO THE CURRENT PAPER. PLEASE NB THE 1ST DIFFERENCED CRFA RESULTS WHICH SHOW THAT THIS EXTENDED SEASON IS STILL SEEN FOR THE FEB-AUGUST SEASON.

L216 – Just an observation, it is a little surprising that RW agrees more strongly with DB and LWB than DB does with LWB. THE PRIMARY AUTHOR HAS NOTED THIS FOR MULTIPLE SPECIES/LOCATIONS THAT DB OFTEN SHOWS SIMILARITIES TO RW, BUT WITH LOWER 1ST ORDER AUTOCORRELATION. REMEMBER THAT ALL OF THESE PARAMETERS SHOULD BE EXPRESSIONS OF SUMMER TEMPERA-TURES SO THEY SHOULD SHOW SOME SIMILARITY. THIS ISSUE NEEDS SPE-CIFIC ATTENTION AT SOME LATER STAGE BUT IS BEYOND THE PRIMARY AIM OF THIS CURRENT PAPER.

L219-220 - Maybe worth clarifying here that EWB would not be expected to contain an actual climate signal in the first place. I DO NOT THINK WE KNOW THAT SPECIFI-CALLY AND WORK BY REVIEWER 1 FOCUSSING ON THE NH SCHWEINGRUBER WORK CLEARLY SHOWS A RELATIONSHIP BETWEEN MINIMUM EARLYWOOD DENSITY AND SPRING TEMPERATURES. IT IS HOWEVER, NOT AS STRONG AS MXD FOR LATE SUMMER TEMPERATURES.

L241-242 – Is it possible that the failed validation could be related to the quality of instrumental data for this early period? THAT IS INDEED LIKELY AND WE CAN EXPAND ON THE DISCUSSION IN THIS REGARD – BUT SEE PREVIOUS COMMENTS.

L278-280 - This may simply indicate that the use of LWB and EWB to calculate DB is an imperfect procedure. It would be necessary to look at data from other locations (and also other species) to identify whether the DB trends in Figure S4 actually reflect inherent properties related to age (and should therefore be detrended) or are related to other factors. This is an important issue and it is unfortunate that this paper does not or cannot investigate this type of issue in more detail. But perhaps at least a bit more discussion could be included in relation to this? WE LIKE TO THINK THAT THIS PAPER HAS EXPLORED THIS ISSUE. THERE IS A CLEAR BIAS IN THE LWB DATA WHICH APPEARS PARTLY (COMPLETELY?) MINIMISED IN THE USE OF THE DB VARIANT. YES – CLEARLY THIS NEEDS MORE TESTING, BUT AT THE SAME TIME, THE RESULTS MAY BECOME LESS AMBIGUOUS AS MORE DATA ARE ADDED.

L294-295 - The results in Figure S4 already indicate that a linear detrending curve can lead to a serious underestimation in the early parts of the series so the poor performance of LINsf is not surprising. YES – BUT THE LINRES IS FINE, SO I THINK WE ARE SEEING AN ISSUE OF THE SF APPROACH RATHER THAN MIS-FITTING OF A DETRENDING FUNCTION PER SE.

L314-317 - But as discussed in the text (and in relation to the results in lines 286-291) it is apparent that LWB is inherently biased. So why even consider it as a feasible option? BECAUSE THIS IS A PAPER EXPLORING THE STRENGTHS AND LIMITATION OF THESE PARAMETERS. SURELY WE NEED TO SHOW THAT THESE DATA ARE BIASED IN THE LOW FREQUENCY DOMAIN. THE HIGH FREQUENCY SIGNAL IS HOWEVER ARGUABLY BETTER THAN DB AND DON'T FORGET THAT LWB DOES IN FACT PASS MOST VALIDATION TESTS SO THIS IS A TRICKY ISSUE TO ASSESS AND OTHER PROXY TEMPERATURE ARCHIVES ARE FUNDAMENTAL IN

[Figure]

FURTHER ASSESSMENT OF THE PRE-1850 TRENDS.

L330 – Maybe include '(Figure 6 and 7)' in the bracket since this reconstruction appears in both figures and is discussed in relation to Figure 7 in the previous paragraph. YES – CAN EASILY ADD THIS IN.

L329-331 – It may be worth stating here something along the lines that the 'best performing' PC and extended reconstructions are shown here and compared with Wiles and the glacial advance record - i.e. state the reason why these reconstructions are shown in Figure 8. YES – CAN EASILY ADD THIS IN.

L358 – Or for species which do not have this colour difference to begin with. YES – CAN EASILY ADD THIS IN.

Figure 3 – It is interesting that LWB calibrates and validates more strongly than DB in terms of the strength of the relationship. Why might this be the case? Could this difference be related to replication? NO – REPLICATION IS THE SAME – THIS APPEARS TO BE RELATED TO THE FACT THAT DB PORTRAYS SUMMERS TEMPERATURES SLIGHTLY MORE WEAKLY THAN LWB. EWB HAS A WEAK CLIMATE SIGNAL AND MAY IMPACT DB WHEN IT IS CALCULATED. THIS CAN BE CLARIFIED FURTHER IN THE PAPER AND DOES LEAD TO THE POSSIBILITY THAT THE BAND-PASS APPROACH TO CALIBRATION USED BY RYDVAL ET AL. (2016/17) COULD BE A REALISTIC APPROACH TO DENDROCLIMATIC RECONSTRUCTION IN THIS REGION. HOWEVER, THIS WAS NOT THE FOCUS OF THIS PAPER WHICH WANTED TO HIGHLIGHT THE STRENGTH AND WEAKNESSES OF THE DIFFERENT PARAMETERS. WE CAN ADD MORE DISCUSSION ON THIS ISSUE.

Figure 6 - It is somewhat difficult to identify the trend of the LINres curve for LWB and especially DB - is it possible to improve the visibility of these curves? Also, in the figure caption (L596-597) maybe consider changing 'low plots' to 'lower set of plots'. AS MANY OF THE TIME-SERIES ARE VERY SIMILAR, IT IS REALLY NOT POSSIBLE TO ADDRESS THE FIRST POINT IN ANY MEANINGFUL WELL. HOWEVER, WE

ARE HAPPY TO EDIT THE CAPTION ACCORDINGLY.

Figure 7 – Why not also show r2 rather than r for the 1850-1900 period? WE CAN DO THIS IF REQUIRED BUT IT DOES NOT CHANGE ANYTHING W.R.T. INTERPRETATION.

Table 1 - Is there any real meaning in including N-EPS information for EWB data? Presumably these data do not (or should not) contain any common climatic signal and there would be no point in developing a chronology from these data that would be of much use. WHY WOULD EWB NOT PORTRAY A COMMON SIGNAL? WE DISAGREE WITH THIS COMMENT.

Table 2 – Minor detail, but why not arrange the site order from west to east? Table 3 – Why is EWB positively (though weakly) correlated with DB? COMPLETELY AGREE. THE ORDER CAN BE EASILY CHANGES TO A MORE LOGICAL EAST-WEST ORDER AS ALREADY PRESENTED IN TABLE 3.

Technical corrections: L32 – affecting instead of effecting AGREED – THIS CAN BE CORRECTED WHEN "ALLOWED" TO EDIT THE PAPER FOR FINAL PUBLICATION

L71 – 'as they are a measure of ...' instead of 'as they measure' may be a more accurate statement. AGREED – THIS CAN BE TWEAKED WHEN "ALLOWED" TO EDIT THE PAPER FOR FINAL PUBLICATION

L109 – reconstructions instead of reconstruction AGREED – THIS CAN BE CORRECTED WHEN "ALLOWED" TO EDIT THE PAPER FOR FINAL PUBLICATION

L126 – Please specify the exact calibration target type as there are different versions. IT8.7/1 is a transparency target whereas IT8.7/2 is a reflective target. I assume that the latter was used. YES - IT8.7/2 – TEXT CAN EASILY BE EDITED.

L211-212 – Consider rewording 'potentially optimal' to something along the lines of 'more optimal compared to a PC approach'. WE PREFER THE CURRENT WORDING AS A PC APPROACH WOULD NOT BE POSSIBLE WITH A SUB-FOSSIL EXTEN-

SION FROM ONE LOCATION.

L218 – Consider specifically stating that the correlation between EWB and LWB is not significant. IN ACTUAL FACT THE CORRELATION IS SIGNIFICANT – BUT WEAK. CURRENT WORDING IS PREFERRED.

Also, 'of' missing in 'the utilization DB to'. REVIEWER 1 ALREADY FLAGGED THIS SENTENCE AS BEING CLUNKY. WE CAN REVISE.

L288 – change 'particular' to 'particularly' AGREED – THIS CAN BE CORRECTED WHEN "ALLOWED" TO EDIT THE PAPER FOR FINAL PUBLICATION

L637 – Is there a better word than 'dominant' which could be used here? CAN BE CHANGED TO "CALIBRATION EXPERIMENTS FOR THE FOUR STRONGEST SEASONS…."
* * *
**TR series replication**

EWB

LWB

DB

reflectance intensity (0-256 scale/1000)

no. of TR series

cambial age-aligned years

**Fig. 1.**